# RLFR: EXTENDING REINFORCEMENT LEARNING FOR LLMS WITH FLOW ENVIRONMENT

## ABSTRACT

Reinforcement Learning with Verifiable Rewards (RLVR) has recently emerged as a promising framework for improving reasoning abilities in Large Language Models (LLMs). However, policy optimized with binary verification prone to overlook potential valuable exploration in reasoning trajectory. In view of heavy annotation cost of golden Process Reward Models (PRMs), recent works attempt using auxiliary signals for reward shaping of process tokens, involving entropy and likelihood collected from logit space. In this work, we offer a novel perspective on shaping RLVR with flow rewards derived from latent space, and propose **RLFR**, where the flow fields of model latents are constructed from either off-policy high-quality data and on-policy rejection sampling data, and the velocity deviations of policy latents within it are quantified to serve as a reward signal. **RLFR** first demonstrates that a well-established flow field can be a sound environment for reward signal collection, highlighting the expressive latent space is much underexplored. Moreover, **RLFR** is able to compress any off-policy expert data as reference for constituting reward signals, and we show that the efficient context dependence compressed within the hidden states are utilized, rather than individual token-level denotation for context comprehending. Experiments on both language and multimodal reasoning benchmarks demonstrate the reliability of flow rewards, and suggesting a promising paradigm for reward shaping with auxiliary signals.

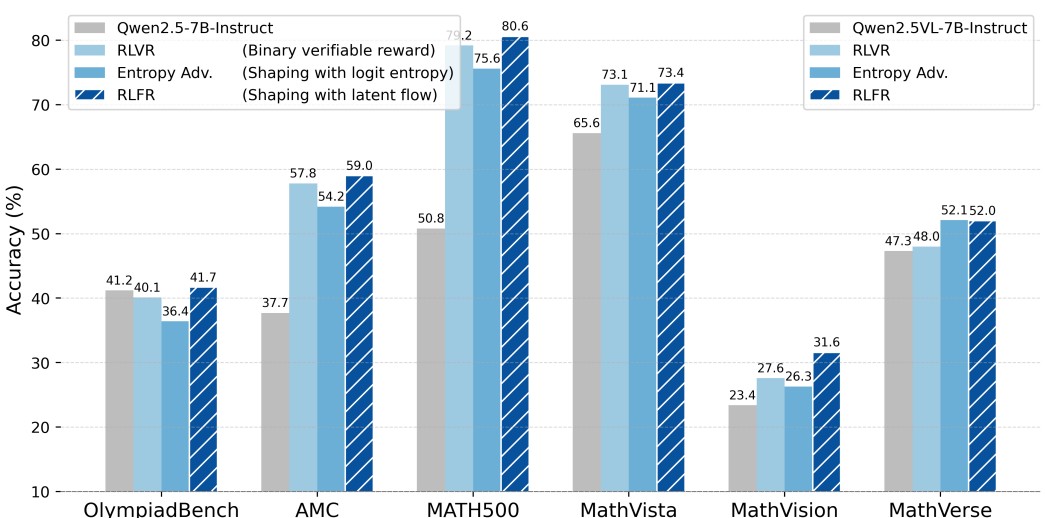

Figure 1: Overall performance on language (left) and multimodal (right) reasoning benchmarks. By introducing flow reward from latent space, RLFR shows consistent progress over RLVR with binary verification and entropy based shaping method (Cheng et al., 2025) from logit space, highlighting the expressive latent space is much underexplored for reward signal collection.

## 1 INTRODUCTION

Recent advances in improving reasoning abilities of Large Language Models (LLMs) underscore the substantial promise of Reinforcement Learning with Verifiable Rewards (RLVR) (Lambert et al.,

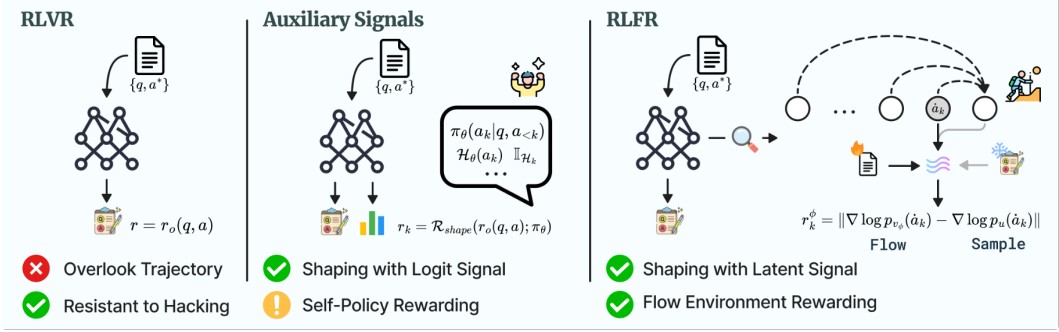

Figure 2: Policy optimized with RLVR prone to overlook potential valuable explorations in reasoning trajectories. To beyond binary verification, auxiliary signals are used for reward shaping of process tokens, involving token entropy and likelihood collected from logit space, where self-policy rewarding risks are non-negligible. Alternatively, we show that the latent space is much underexplored yet highly expressive and a well established flow field can be a sound environment for yielding flow reward from velocity deviations and extending RLVR with latent reward utilization.

2024; Jaech et al., 2024; DeepSeek-AI et al., 2025). By incentivizing the optimization of LLMs with outcome verification, there are far less susceptible to reward hacking. However, the binary verification prone to overlook potential valuable policy explorations in reasoning trajectories in cases that the answers are difficult to derive with part of correct trajectory (Hammoud et al., 2025), thus provides an intolerant reward signal with decreased exploratory behavior (Cui et al., 2025b).

A more natural way to address these issues is to provide step-by-step process rewards along reasoning trajectories with Process Reward Models (PRMs) (Zhao et al., 2025; Liu et al., 2025d; Yang et al., 2025a), however, the heavy annotation costs of intermediate steps pose a significant bottleneck for scalability at time, and the misalignment between PRMs training corpora and online reasoning trajectories further introduce reward gaps (Ye et al., 2025). Alternatively, the value model in PPO framework (Schulman et al., 2017; Yue et al., 2025; Yuan et al., 2025) offers a promising strategy, but the effective credit assignment over the binary outcome reward are still underexplored.

To encouraging policy exploration, recent works leverage auxiliary signals for reward shaping (Ng et al., 1999) beyond binary outcome verification, involving model likelihood (Damani et al., 2025; Li et al., 2025b; He et al., 2025) and token entropy (Cheng et al., 2025; Wang et al., 2025a) collected from logit space. While the confidence may serve as an appropriate indicator for examining policy states, it may not be well-suited for constituting reward signals for optimization (Wang et al., 2025b; Cui et al., 2025b). As the self-policy rewarding may cause LLM over exploits its own confidence estimates rather than learning genuinely improved reasoning strategies, where the potential hacking risks are non-negligible, and may undermine prolong RL training (Liu et al., 2025b;c).

In this work, we propose **RLFR**, that offering a novel framework on shaping RLVR with flow rewards for regarding reasoning trajectory. We aspire to explore *whether the broader latent space of LLMs encompass productive signals for reward utilization with reliable stability*. RLFR first constructs flow fields (Lipman et al., 2022; Liu et al., 2022) of model latents from either off-policy high-quality data and on-policy rejection sampling data, and the velocity deviations of policy latents within it are quantified to serve as a reward signal. While the larger deviations are penalized as drifting away from the reference distribution formed by flow, and smaller deviations are encouraged. The flow fields are online updated alongside the policy optimization, with rejection sampling data filtered by desired metrics. We also formally show that the evidence lower bound of log-likelihood is constituted by negative velocity deviation, thus establishing the connection between velocity deviation and probability likelihood with inverse correlation under reference distribution.

Particularly, RLFR first demonstrates that a well-established flow field can be a sound environment for reward signal collection, yielding stable performance gains throughout RL training with no sign of degeneration. And we also highlight that the expressive latent space are highly underexplored as a substrate for reward design, complementing prior auxiliary signals from logit space. Moreover, RLFR provides a natural way to leverage expert reasoning trajectories from off-policy data into the constitution of reward signals through reference flow fields. Experiments on both language and

multimodal reasoning benchmarks across Qwen and Llama models demonstrate the reliability of flow rewards with consistent progress over baseline RLVR methods.

The contributions of this work can be summarized as follows: (1) We propose RLFR, a promising reward shaping framework with flow rewards derived from LLMs latents, extending RLVR with latent rewards utilization. (2) We demonstrate that a well-established flow field can be a sound environment for reward signal collection, highlighting the expressive latent space is much underexplored. (3) Both off-policy expert data and on-policy rejection sampling data are introduced for constituting reward signals as flow reference . (4) Comprehensive experiments on diverse reasoning benchmarks across both language and multimodal models demonstrate the reliability of our framework. All the codes, data, and model weights are released to foster future research in this area.

## 2 PRELIMINARIES

### 2.1 REINFORCEMENT LEARNING WITH VERIFIABLE REWARDS

Reinforcement Learning with Verifiable Rewards (RLVR) (Lambert et al., 2024) constitutes a general post-training paradigm where model response can be deterministically verified. Let $\pi_\theta$ be an LLM parameterized by $\theta$, that receives a prompt $q$ and generates a token sequence $\boldsymbol{a} = (a_1, ..., a_K)$ as response. A binary verifier then assigns a scaler reward $r_o(q, \boldsymbol{a}) \in \{0, 1\}$ to each prompt-response pair, where $r_o$ underlines its outcome nature. The goal of RL is to maximize the expected reward:

$$\mathcal{J}(\theta) = \mathbb{E}_{q\sim\mathcal{D},\boldsymbol{a}\sim\pi_\theta(\cdot|q)}[r_o(q, \boldsymbol{a})]. \tag{1}$$

Here, $\mathcal{D}$ is a dataset of prompts with corresponding ground-truth answer. Despite robustness against reward hacking, the coarse granularity outcome rewards make RLVR prone to overlooking the potential valuable policy exploration in reasoning trajectories in cases of derived incorrect answers.

**RLVR Algorithms.** Group Relative Policy Optimization (GRPO) (Shao et al., 2024) as a widely used reinforcement learning algorithm that simplfies the Proximal Policy Optimization (PPO) (Schulman et al., 2017) by discarding the value model for baseline advantage estimation. Sampling a group of response $\{\mathbf{a}_i\}_{i=1}^G$ per prompt and using their average reward as baseline, and the clipped surrogate objective is preserved as PPO, leading to the following maximization objective:

$$\mathcal{J}_{\text{GRPO}}(\theta) = \mathbb{E}_{q\sim\mathcal{D}, \{\boldsymbol{a}_i\}_{i=1}^G\sim\pi_\theta(\cdot|q)} \left[ \frac{1}{G}\sum_{i=1}^G \frac{1}{|a_i|} \sum_{t=1}^{|a_i|} \min\left(\rho_{i,k}\widehat{A}_i, \text{clip}\left(\rho_{i,k}, 1-\varepsilon, 1+\varepsilon\right)\widehat{A}_i\right)\right], \tag{2}$$

where $\rho_{i,k} = \frac{\pi_\theta(a_{i,k}|q,a_{i,<k})}{\pi_{\theta_{\text{old}}}(a_{i,k}|q,a_{i,<k})}$ is the importance sampling ratio between the current and old policy models, and the advantage $\widehat{A}_{i,o}$ is shared among all tokens within response $\boldsymbol{a}_i$ and is computed as

$$\widehat{A}_{i,o} = \frac{r_{i,o} - \text{mean}(\{r_{i,o}\}_{i=1}^G)}{\text{std}(\{r_{i,o}\}_{i=1}^G)}. \tag{3}$$

We denote $r_{i,o} = r_o(q, \boldsymbol{a}_i)$ for simplicity and clear comparison with later introduced dense rewards.

**Reward Shaping.** As a common technique in reinforcement learning for accelerating and stabilizing policy optimization, reward shaping transforms explicit environment-based rewards into a proxy reward function, which typically involves operations like clipping or shifting, *etc.*, (Wang et al., 2024c), and may also incorporate auxiliary signals, such as response length or token entropy to steer the model toward desired behavior more effectively (Arora & Zanette, 2025; Cheng et al., 2025).

### 2.2 FLOW MATCHING

Flow Matching (FM) (Lipman et al., 2022; Liu et al., 2022) defines a generative process that learns a continuous-time velocity field, transporting samples from a simple prior distribution $p_{\text{init}}$ (e.g., Gaussian) into the target data distribution $p_{\text{data}}$. With data pairs $(\dot{\boldsymbol{x}}_0, \dot{\boldsymbol{x}}_1)$ sampled from $p_{\text{init}} \times p_{\text{data}}$, the forward process is given by the linear interpolation: $\dot{\boldsymbol{x}}_t = (1-t)\dot{\boldsymbol{x}}_0 + t\dot{\boldsymbol{x}}_1$, and the neural network $\boldsymbol{v}_\phi$ is trained to predict the target velocity field $\boldsymbol{u} = \dot{\boldsymbol{x}}_1 - \dot{\boldsymbol{x}}_0$ by minimizing the flow matching loss:

$$\mathcal{L}_{\text{FM}}(\dot{\boldsymbol{x}}_t; \phi) = \mathbb{E}_{t\sim\mathcal{U}[0,1],\dot{\boldsymbol{x}}_0\sim p_{\text{init}},\dot{\boldsymbol{x}}_1\sim p_{\text{data}}} \left[\|\boldsymbol{v}_\phi(\dot{\boldsymbol{x}}_t, t) - (\dot{\boldsymbol{x}}_1 - \dot{\boldsymbol{x}}_0)\|^2\right]. \tag{4}$$

We denote $\dot{\boldsymbol{x}}$ as latent signal and distinguish it from tokens. The $\boldsymbol{v}_\phi$ characterizes the $p_{\text{data}}$ with the flow field through accurate velocity prediction, where the outsider can be notably identified.

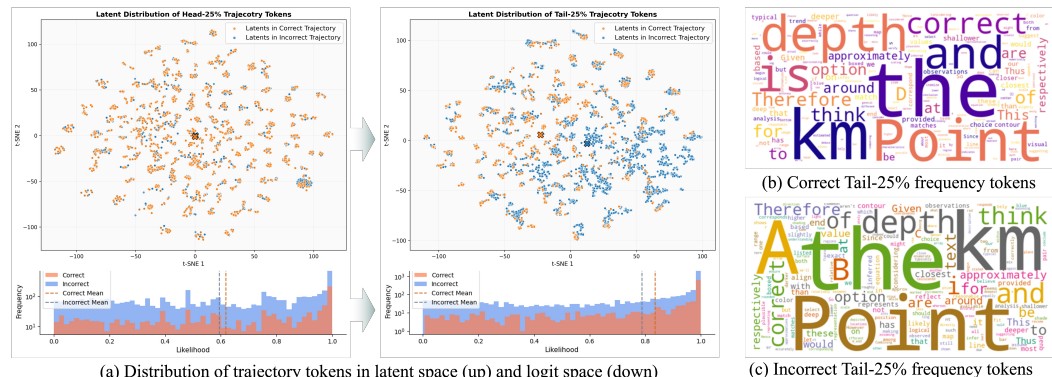

(b) Correct Tail-25% frequency tokens

(a) Distribution of trajectory tokens in latent space (up) and logit space (down)

(c) Incorrect Tail-25% frequency tokens

Figure 3: Distribution of trajectory tokens in LLM reasoning. **(a) Distribution of trajectory tokens in latent space (up) and logit space (down).** We perform 256 rollouts for prompt randomly sampled from MATH (Hendrycks et al., 2021). The latent distribution show progressively expressive signals on tail trajectory tokens, as continuously interacting with preceding tokens for context compression. In contrast, neither the logit distribution nor the **(b) & (c) textual clouds of reasoning trajectories** reveal any distinguishable signals, highlighting the potential of latent space for reward utilization.

## 3 RLFR

In this section, we first show that the latent space is highly expressive for reward utilization, and then introduce the flow reward formed by velocity deviation, where the connection with probability likelihood are established. We then introduce RLFR, that extends RLVR with flow rewards for process tokens in advantage estimation, while the flow conditions are involved to introduce context dependency. The flow fields are online updated with rejection sampling data during policy optimization.

### 3.1 ANALYZING LATENT SPACE IN REASONING TRAJECTORY

Previous studies have shown the effectiveness of signals from logit space in shaping binary verification reward (Damani et al., 2025; Cheng et al., 2025; Wang et al., 2025b; Yu et al., 2025b), while the broader latent space is much underexplored. In light of this, we further analyze the latents of reasoning trajectory tokens to evaluate the reliability of latent space for reward utilization. We use Qwen2.5-7B (Yang et al., 2024) to generate 256 rollouts for prompt randomly sampled from MATH (Hendrycks et al., 2021) with 0.7 temperature for decoding, and ultimately get $10^5$ response tokens within the same subject content along with their corresponding latents. The latents are extracted from layers $\mathcal{L} = (\tau_0 L, ... \tau_N L)$ for probing, where $L$ denotes the total number of model layers, and $\tau_i \in [0, 1]$ specifies the percentile positions for extraction. We show the case of $\tau_i = 0.5$ in Figure 3, while additional results are provided in Appendix B and show the similar tendency.

**The latents of tail tokens in reasoning trajectory show progressively pronounced signals when distinguished by answer correction.** We observe that compared to head tokens in reasoning trajectory where basically no identifiable signals are emerged, the latents of tail tokens are much more expressive as continuously interacting with all preceding tokens for context compression. However, neither the logit space likelihood nor the textual clouds exhibit any noticeable signals.

**Subset of latents in reasoning trajectory with incorrect answer exhibit close to those in correct trajectories, while RLVR with outcome verification penalizes them.** We discern that penalization should be applied to latents with substantial deviations from high-quality trajectory latents, while similar latents with minor deviations should be treated tolerantly.

### 3.2 FLOW REWARD FROM VELOCITY DEVIATION

Encouraged by the expressive latent space, a sound yet efficient metric is required to collect emerging signals for reward utilization, and we consider flow matching as underlying framework for which typically used in continuous modeling with established velocity field. Inspired by (Li et al., 2025c), we notice that instead of using predicted velocity to reverse the forward process for data distribution

generation, the accuracy of velocity prediction can serve as a sensible metric to evaluate whether current samples are within the data distribution formed by flow. And we rewrite the Eq. 4 as

$$\mathcal{R}_{\text{FM}}^{\phi}(\dot{\boldsymbol{a}}_k; t, \tau) = \|\boldsymbol{v}_\phi(\dot{\boldsymbol{a}}_{k,t}, t) - (\dot{\boldsymbol{a}}_{k,1} - \epsilon)\|^2, \quad \text{where } t, \tau \sim \mathcal{U}[0,1], \ \epsilon \sim \mathcal{N}(0, I), \quad (5)$$

where $\dot{\boldsymbol{a}}_k$ denotes latents of token $\boldsymbol{a}_k$, and $\dot{\boldsymbol{a}}_{k,t}$ is the linear interpolation between $\dot{\boldsymbol{a}}_k$ and $\epsilon$. The flow network $\boldsymbol{v}_\phi$ is first pre-trained on latents of high-quality data to establish the reference distribution for offline start, and then frozen for flow reward calculation that evaluates the velocity deviation of the current sample $\dot{\boldsymbol{a}}_k$ under the reference flow field.

**Debiasing the Timestep Impacting.** The flow network $\boldsymbol{v}_\phi$ provides velocity drifts toward reference distribution through the whole reverse process, which underscores the challenge on timestep priority in deviation evaluation for flow reward. Considering the connection between velocity prediction and score function (Gao et al., 2024; Liu et al., 2025a), we present the score given by $\boldsymbol{v}_\phi$ as:

$$\nabla_{\dot{\boldsymbol{a}}_{k,t}} \log p_{\boldsymbol{v}_\phi}(\dot{\boldsymbol{a}}_{k,t}) = -\frac{\dot{\boldsymbol{a}}_{k,t}}{1-t} + \frac{t}{1-t} \boldsymbol{v}_\phi(\dot{\boldsymbol{a}}_{k,t}, t). \quad (6)$$

A detailed proof is provided in Appendix C.1. Instead of relying on global-consistent direction provided by velocity prediction, the score function provides more accurate drift direction from local distributional gradients, which is more practical for deviation evaluation, and we have

$$\mathcal{R}_{CFM}^{\phi}(\dot{\boldsymbol{a}}_k; \mathcal{T}, \mathcal{L}) = \mathbb{E}_{t \sim \mathcal{T}, \tau \sim \mathcal{L}} \left[ \|\nabla_{\dot{\boldsymbol{a}}_{k,t}} \log p_{\boldsymbol{v}_\phi}(\dot{\boldsymbol{a}}_{k,t}) - \nabla_{\dot{\boldsymbol{a}}_{k,t}} \log p_{\boldsymbol{u}}(\dot{\boldsymbol{a}}_{k,t})\|^2 \right]$$
$$= \mathbb{E}_{t \sim \mathcal{T}, \tau \sim \mathcal{L}} \left[ \frac{t}{1-t} \mathcal{R}_{FM}^{\phi}(\dot{\boldsymbol{a}}_k; t, \tau) \right], \quad (7)$$

where $\mathcal{R}_{CFM}^{\phi}$ is used for calculating flow reward with timestep collection $\mathcal{T}$ by debiasing weighting. We suggest that the velocity deviation serves as a surrogate for score deviation, while the coefficient emphasizes the timestep priority through the reverse process. We provide ablations in Sec. 4.3 that using flow reward at different timesteps for RL, where the larger timesteps with less noises are favorable in constituting reward signal, which is consistent with suggestion given by Eq. 7.

**Theoretical Analysis with Likelihood.** Comparing to probability likelihood under reference distribution, velocity deviations show directional drifts error toward reference distribution. These two paradigms appear as different profile for distribution evaluation, and we further clarify their underlying relationship as:

$$\log p_{\boldsymbol{v}_\phi}(\dot{\boldsymbol{a}}_k) \geq C(\dot{\boldsymbol{a}}_k) - \lambda \mathbb{E}_{t, \tau \sim \mathcal{U}[0,1]} \left[ \mathcal{R}_{\text{FM}}^{\phi}(\dot{\boldsymbol{a}}_k; t, \tau) \right], \quad (8)$$

where $\log p_{\boldsymbol{v}_\phi}(\dot{\boldsymbol{a}}_k)$ is the log-likelihood under distribution parameterized by $\boldsymbol{v}_\phi$, $\lambda > 0$ is constant, and $C(\dot{\boldsymbol{a}}_k)$ is the sundry term. We show that the evidence lower bound of log-likelihood is constituted by negative velocity deviation under reference distribution, indicated that the two paradigms are inversely correlated, and the minimal $\mathcal{R}_{\text{FM}}^{\phi}(\dot{\boldsymbol{a}}_k; t, \tau)$ with respect to a given $\dot{\boldsymbol{a}}_k$ corresponds to the maximal evidence lower bound (ELBO) on the log-likelihood $\log p_{\boldsymbol{v}_\phi}(\dot{\boldsymbol{a}}_k)$. We calibrate the sign of flow rewards in RLFR, and provide the detailed proof of Eq. 8 in Appendix C.2.

### 3.3 EXTENDING RLVR WITH FLOW REWARD

**Velocity-Based Advantage Shaping.** The main idea behind RLFR is to leverage the expressive latent space of LLMs with flow reward and thus extend RLVR with latent rewards utilization. Instead of sharing common advantage within response $\boldsymbol{a}$, we shape advantage term for each token $\boldsymbol{a}_k$ with flow returns, yielding by the accumulation of decayed flow rewards. While the advantage shaping makes it more flexible for different RLVR algorithms, without considering the specific advantage estimation methods. We have:

$$\hat{A}_k = \sum_{s=k}^{|\boldsymbol{a}|} \gamma^{s-k} r_s^{\boldsymbol{v}_\phi} + \hat{A}_o, \quad (9)$$

$$r_k^{\boldsymbol{v}_\phi} = -\beta \cdot \hat{r}_k^{\boldsymbol{v}_\phi} \mathbb{I}[|\hat{r}_k^{\boldsymbol{v}_\phi}| > \eta], \quad \text{where } \hat{r}_k^{\boldsymbol{v}_\phi} = \text{minmax}(\{\mathcal{R}_{CFM}^{\phi}(\dot{\boldsymbol{a}}_k); \mathcal{T}, \mathcal{L}\}_{k=1}^{|\boldsymbol{a}|}), \quad (10)$$

where $\mathbb{I}[\cdot]$ is the indicator function and return 1 if condition is true and 0 otherwise, where we discard noisy fluctuations in flow rewards and preserve only substantial deviations above $\eta$. We perform the

---

**Algorithm 1:** Reinforcement Learning with Flow Rewards (RLFR)

---

**Inputs:** Online data $\mathcal{Q}\{q, a\}$, offline data $\mathcal{D}_{off}\{q, z, a\}$, reference data buffer $\mathcal{B}$, initial flow model $\boldsymbol{v}_\phi$, validated batch size $\kappa$, layer collection $\mathcal{L} = (l_1, ..., l_N)$, response quality metric$(\cdot)$

**Offline Start:**

Extract policy latents from layer collection $\mathcal{L}$ on $\mathcal{D}_{off}$ and construct the $\mathcal{D}_{off}^*$

Perform flow training on $\mathcal{D}_{off}^*$ with Eq. 4 where the loss is only calculated on response tokens

**Online Optimization:**

Initialize reference data buffer $\mathcal{B} \leftarrow \emptyset$

**while** Training **do**

    Generate rollouts $\mathcal{G}$ for batch data from $\mathcal{Q}$

    Optimize the policy with RL algorithms such as Eq. 2 with advantage estimated for each token by Eq. 9

    $\mathcal{B} \leftarrow$ Rejection-Sampling$(\mathcal{G}, \text{metric})$ ;    // Recommend metrics: correctness, entropy

    **while** $|\mathcal{B}| > \kappa$ **do**

        Optimize flow $\boldsymbol{v}_\phi$ on batch data from $\mathcal{B}$ with policy latents in $\mathcal{L}$ using Eq. 4

        $\mathcal{B}$.pop(batch data)

    **end while**

**end while**

**return** policy, flow $\boldsymbol{v}_\phi$ and end training

---

minmax-norm within the single sequence to regularize the numerical values between $[-1, 1]$. $\mathcal{T}$ and $\mathcal{L}$ are the collections of timesteps and layer percentiles used to calculate the velocity deviations, and the latents are detached from the computational graph for stopping backpropagation. Practically, we incorporate the latents of subsequent token $\hat{\boldsymbol{a}}_{k+1}$ to serve as conditions for assisting velocity prediction in flow reward, that further establishes context dependence with enlarged interactive space, and more ablations are provide in Sec 4.3. The flow reward provides a stable examination on model latents that quantify their velocity deviation from flow pre-trained on off-policy high-quality data.

**Updated Rewards with Rejection-Sampling.** As the policy are progressing during optimization alongside with their latents (Huan et al., 2025), yielding flow rewards from frozen $\boldsymbol{v}_\phi$ pre-trained on offline start data introduces inherent distribution gap. Therefore, we update flow by Eq. 4 throughout the policy optimization with online rejection-samping data, where the filtered metrics are flexible to direct the constitution of reference distribution for flow reward calculation. We provide detailed framework in Algorithm 1, and we empirically found that the correctness is still the most effective metric, where more ablations are provided in Sec. 4.3 for comparison.

## 4 EXPERIMENTS

### 4.1 EXPERIMENTAL SETUP

**Training Data.**

We conduct experiments for both language and multimodal models for evaluation. In language settings, we use openr1 (Hugging Face, 2025) as offline start data for flow pretraining, which contains 93k carefully curated mathematical reasoning problems. The reinforcement learning of RLFR is performed on MATH (Hendrycks et al., 2021), which includes diverse reasoning-intensive problems spanning algebra, geometry, number theory, and combinatorics. In multimodal settings, we filter the math subset from MMPR (Wang et al., 2024b) as offline data for flow pretraining, which consists 115k multimodal mathematical reasoning problems. Subsequently, the reinforcement learning is conducted on the MMK12 (Meng et al., 2025a), which includes mathematics, physics, and general science with multimodal contexts.

**Evaluation.** We assess the language reasoning performance using a suite of standard mathematical reasoning benchmarks, including: AIME24/25 [1], AMC23 [2], MATH500 (Hendrycks et al., 2021), and OlympiadBench (He et al., 2024). We report Pass@1 metric with rollout temperature of 0 and Pass@32 with temperature of 0.7 for decoding, under a maximum response length of 8192 tokens. For multimodal reasoning benchmarks, we include MathVista (Lu et al., 2023), MathVerse (Zhang et al., 2024), MathVision (Wang et al., 2024a), WeMath (Qiao et al., 2024), and logic benchmarks

---

[1]https://huggingface.co/datasets/AI-MO/aimo-validation-aime

[2]https://huggingface.co/datasets/AI-MO/aimo-validation-amc

Table 1: Overall performance on language reasoning benchmarks. Pass@32 and Pass@1 metrics are reported with zero-shot evaluation. † means the model trained in our setting for evaluation.

| Model | AIME25 | | AIME24 | | AMC23 | | MATH500 | | OlympiadBench | |
|---|---|---|---|---|---|---|---|---|---|---|
| | Pass@32 | Pass@1 | Pass@32 | Pass@1 | Pass@32 | Pass@1 | Pass@32 | Pass@1 | Pass@32 | Pass@1 |
| *Qwen2.5-Math-1.5B* | 26.7 | 0.0 | 43.3 | 13.3 | 79.5 | 32.5 | 90.2 | 31.8 | 61.9 | 22.8 |
| RLVR | 23.3 | 3.3 | 46.7 | 16.7 | 81.9 | 41.4 | 91.8 | 71.0 | 62.7 | 33.5 |
| **RLFR** | 30.0+6.7 | 6.7+3.4 | 50.0+3.3 | 13.3-3.4 | 83.1+1.2 | 44.6+3.2 | 92.4+0.6 | 72.0+1.0 | 63.1+0.6 | 35.7+2.2 |
| *Qwen2.5-Math-7B* | 30.0 | 3.3 | 56.6 | 16.6 | 80.7 | 37.3 | 94.2 | 50.8 | 61.2 | 17.2 |
| Qwen2.5-Math-7B-Inst | 36.6 | 10.0 | 46.6 | 13.3 | 79.5 | 50.6 | - | 79.8 | - | 41.2 |
| Oat-Zero | 30.0 | 6.7 | 56.6 | 30.0 | 81.9 | 55.4 | - | 79.6 | - | 42.6 |
| PRIME | 33.3 | 13.3 | 53.3 | 16.7 | 79.5 | 55.4 | 91.8 | 76.2 | 65.3 | 37.9 |
| RLVR | 30.0 | 10.0 | 56.6 | 26.7 | 80.7 | 57.8 | 92.0 | 79.2 | 64.8 | 40.1 |
| Entropy Adv.† | 26.6 | 6.6 | 53.3 | 20.0 | 75.9 | 54.2 | 93.0 | 75.6 | 61.2 | 36.4 |
| **RLFR** | 33.3+3.3 | 10.0+0.0 | 56.6+0.0 | 30.0+3.3 | 83.1+2.4 | 59.0+1.2 | 92.6+0.6 | 80.6+1.4 | 66.1+1.3 | 41.7+1.6 |
| *Llama3.1-8B* | 0.0 | 0.0 | 0.0 | 0.0 | 14.4 | 0.0 | 28.2 | 0.6 | 5.9 | 0.0 |
| Llama3.1-8B-Inst | 16.7 | 3.3 | 26.7 | 10.0 | 69.8 | 21.7 | - | 48.0 | - | 14.7 |
| RLVR | 6.7 | 0.0 | 6.7 | 0.0 | 25.3 | 7.2 | 46.0 | 18.0 | 18.6 | 4.0 |
| **RLFR** | 13.3+6.6 | 6.7+6.7 | 16.7+10.0 | 6.7+6.7 | 27.7+2.4 | 12.0+4.8 | 46.4+0.4 | 18.8+0.8 | 19.3+0.7 | 11.3+7.3 |

Table 2: Overall performance on multimodal reasoning benchmarks. **Logic Avg.** denotes average of LogicVista and VisuLogic. **Math Avg.** denotes average of other four math benchmarks.

| | MathVista | MathVision | MathVerse | WeMath | LogicVista | VisuLogic | **Math Avg.** | **Logic Avg.** | **Avg.** |
|---|---|---|---|---|---|---|---|---|---|
| *Qwen2.5VL-3B-Inst* | 62.0 | 21.1 | 33.7 | 40.7 | 38.9 | 26.8 | 39.4 | 32.9 | 37.2 |
| RLVR | 65.9 | 24.1 | 42.2 | 55.2 | 41.1 | 26.4 | 46.8 | 33.7 | 42.5 |
| **RLFR** | 67.7+1.8 | 29.6+5.5 | 42.6+0.4 | 56.9+1.7 | 42.3+1.2 | 25.4-1.0 | 49.2+2.4 | 33.8+0.1 | 44.1+1.6 |
| *Qwen2.5VL-7B-Inst* | 65.6 | 23.4 | 47.3 | 53.4 | 47.8 | 27.1 | 47.4 | 37.5 | 44.1 |
| R1-OneVision-7B | 63.7 | 22.4 | 45.2 | 52.9 | 38.9 | 18.3 | 46.1 | 28.6 | 40.2 |
| OpenVLThinker-7B | 64.5 | 24.3 | 46.1 | 50.3 | 38.7 | 10.6 | 46.3 | 24.6 | 39.1 |
| MM-Eureka-7B | 73.5 | 27.9 | 51.9 | 58.7 | 46.9 | 25.5 | 53.0 | 36.2 | 47.4 |
| RLVR | 73.1 | 27.6 | 48.0 | 64.6 | 48.3 | 24.8 | 53.3 | 36.5 | 47.7 |
| Entropy Adv. † | 71.1 | 26.3 | 52.1 | 63.1 | 44.5 | 25.7 | 53.2 | 35.1 | 47.1 |
| **RLFR** | 73.4+0.3 | 31.6+4.0 | 52.0+4.0 | 66.1+1.5 | 48.3+0.0 | 26.7+1.9 | 55.8+2.4 | 37.5+1.0 | 49.7+1.9 |

including LogicVista (Xiao et al., 2024) and VisuLogic (Xu et al., 2025). We adopt greedy decoding with temperature of 0 and report Pass@1 metric for evaluation.

**Implementation Details.** We adopt RL algorithm as GRPO in our experiments. In flow pretraining, we use a training batch size of 128 with $10^{-4}$ learning rate and warmup ratio of 0.1, where the LLM backbone is frozen. We empirically set the percentiles of layer collection as $\{0.25, 0.5, 0.75\}$ throughout the model, where the layer position embedding are added. The flow network comprises 4 layers for 3B models and 6 layers for 7B/8B models. In reinforcement finetuning, we exclude both KL divergence loss and entropy loss, and use a training batch size of 128 with policy learning rate of $10^{-6}$ and flow learning rate of $10^{-4}$, where the maximal response length is set for 4096. We use the threshold $\eta$ of 0.6 to discard noisy fluctuations, the validated batch size $\kappa$ of 32 for flow update, while the discount factor $\gamma$ and coefficient $\beta$ are set as 1 and 0.01. The timestep collection $\mathcal{T}$ for yielding flow reward is set for $\{0.8\}$, and we use the temperature of 1 for response rollout.

## 4.2 MAIN RESULTS

The main experimental results in Table 1 and Table 2 demonstrate that RLFR consistently outperforms baselines across both language and multimodal reasoning benchmarks. The baselines include: basic RLVR with binary verification, existing approaches (Liu et al., 2025e; Yang et al., 2025b; Deng et al., 2025; Meng et al., 2025b), process implicit rewards (Cui et al., 2025a), and entropy-based advantage shaping method (Cheng et al., 2025), which serves as a strong baseline for logit-space comparison. Table 1 reports results on language reasoning benchmarks using Qwen and Llama base models across 1.5B and 7B/8B size, where RLFR shows consistent improvement, surpassing basic RLVR by 1.5% average score on Qwen2.5-Math-7B and by 5.3% average score on Llama3.1-8B, while achieving superior performance compared to entropy-based shaping method in logit space. Table 2 presents results on multimodal reasoning benchmarks, where RLFR achieves compelling improvements on challenging benchmarks like MathVision and MathVerse, and show steady generalization on out-of-domain logic benchmarks. The performance gains across both language and multimodal reasoning benchmarks on different model families show that the flow rewards derived from latent space reliably advance the performance with binary verification, while competitive to logit space shaping.

| Steps | Type | Frequently Reward Tokens |
|-------|------|--------------------------|
| 25 | + | frac theta +-={} sqrt cdot matrix times abs pi math symbol sin line product set angle... |
|  | - | To output of need equation problem value these many how determine import break maximum... |
| 286 | + | frac sqrt area times cdot average triangle sin equality outer second integers length sum... |
|  | - | To problem output determine understand python frac denote Given integers use properties... |

Table 3: Case study of reward tokens in training progress. "+" means positive flow reward and "-" means negative flow reward.

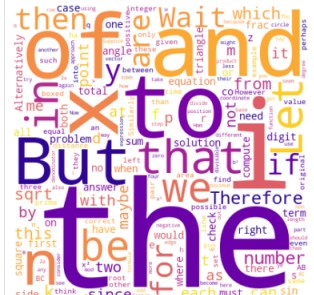

Figure 4: Textual cloud of offline start dataset.

## 4.3 ABLATIONS

We conduct ablations on the framework of RLFR and the variant of flow reward. The ablations are conducted on multimodal settings with Qwen2.5VL-3B, where Pass@1 metric is reported.

**Effect of online rejection sampling and corresponding metrics.** We present the ablations of RLFR framework in Table 4, where both offline start of flow network and online updated with rejection sampling show steady contribution to the final performance. Additionally, noisy fluctuations in flow reward appear to be detrimental and the filtering for substantial deviations is essential for stable RL training. In Table 5, we present the effects of metric choices in rejection sampling that incentivize the desired reference distribution of the flow, including outcome correctness by binary verification, average trajectory entropy above 50th percentile, and their composition, and the correctness is still validated to be the most effective metric on filtering the reference data.

Table 4: Ablation experimental results of model trained under incomplete RLFR framework.

| Method | Math Avg. | Logic Avg. |
|--------|-----------|------------|
| **RLFR** | **49.2** | **33.8** |
| w/o offline start | 45.6-3.6 | 31.3-2.5 |
| w/o rejection sampling | 46.8-2.4 | 32.4-1.4 |
| w/o fluctuation filtering. | 47.6-1.6 | 32.6-1.2 |

Table 5: Ablation results of metrics used in rejection sampling that contribute to the online updated reference distribution for flow reward.

| Metric | Math Avg. | Logic Avg. |
|--------|-----------|------------|
| Outcome correctness | **49.2** | 33.8 |
| Trajectory entropy | 47.9 | **33.9** |
| Correctness+Entropy | 46.7 | 33.1 |

**Effect of timesteps debiasing in flow reward.** The flow reward derived from velocity deviations delivers several impact factors, including timesteps used for velocity prediction and conditions in flow network, and we provide their effect on reward quality in Figure 5 and Table 6. The results in Figure 5 provide strong evidence for timesteps debiasing method suggested in Sec. 3.2, where the larger timesteps with less noisy are preferred for flow reward, as their more reliable velocity prediction, and when using collection of timesteps such as $\mathcal{T} = \{0.2, 0.4, 0.6, 0.8\}$ for calculating velocity deviation, the debiasing weighting further shows superiority by 1.7% average score improvement compared to equal average.

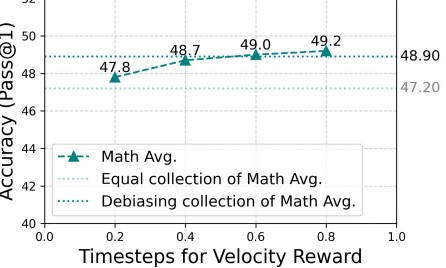

Figure 5: Results on different timesteps for flow reward and debiasing effect.

**Effect of flow conditions in velocity prediction.** We provide results with different velocity prediction conditions for constructing flow reward in Table 6, including latents from identity token, previous token, and post token, where the post token condition exhibits apparent advantage compared to other two paradigms, validating that using larger auxiliary space to support velocity prediction is beneficial for precedent reward quality.

Table 6: Ablations results of auxiliary conditions in flow network for velocity prediction.

| Condition | Position | Math Avg. | Logic Avg. |
|-----------|----------|-----------|------------|
| $\dot{a}_k$ | Identity | 48.9 | **34.0** |
| $\dot{a}_{k-1}$ | Previous | 45.8 | 32.6 |
| $\dot{a}_{k+1}$ | Post | **49.2** | 33.8 |

## 4.4 ANALYSIS

We further analysis the reward behaviors derived from velocity deviations during training progress to better understand what is being encouraged. In Table 3, we found that: **(1)** Contrast to previous entropy-based method that encouraging tokens with logical connection function to dominate reasoning directions (Wang et al., 2025b; Cheng et al., 2025), the flow rewards prefer tokens that practically execute the question, and depress tokens with empty content such as connection tokens. We attribute this to that the high entropy tokens typically correspond to ambiguity hidden states that prepared for a large set of candidate tokens, which makes it hard to predict in flow field. **(2)** The positive reward tokens are initially related to offline start data as shown in Figure 4, such as `sqrt`, `angle`, and progressively updated with policy. **(3)** Despite the large portion of general words in pretrained dataset, the flow yields limited reward on these tokens rather than completely matching. Additionally, some tokens receive either positive reward or negative reward in cases, e.g., `frac`, indicating that the flow reward is capable of relying on efficient context dependence compressed within the hidden states, rather than individual token-level denotation for context comprehending.

> **Takeaways**
>
> - **Flow rewards prefer tokens that practically execute the question**, and depress tokens with empty content such as connection tokens.
> - **High entropy in logit space makes larger velocity deviations in latent space**, attributing to the ambiguity hidden states correspond to a large set of candidate tokens.
> - **Flow rewards rely on efficient context dependence compressed within the hidden states**, rather than individual token-level denotation for context comprehending.

## 5 RELATED WORK

**Reinforcement learning beyond binary verification.** Reinforcement learning with binary verifiable reward has recently demonstrates promising effectiveness in advancing reasoning abilities of Large Language Models (Yu et al., 2025a; Hu et al., 2025; DeepSeek-AI et al., 2025). Despite robustness to reward hacking, the binary verification largely restricts the potential valuable exploration in reasoning trajectory. Indicated by the policy entropy (Cui et al., 2025b; Wang et al., 2025b), recent practices leverage a variety of metrics derived from model likelihood either to shape the reward signal (Cheng et al., 2025; Damani et al., 2025; He et al., 2025; Li et al., 2025b), or to serve as indicators for identifying tokens with different optimization (Wang et al., 2025a;b; Fu et al., 2025). However, this work highlights the latent space could be an expressive substrate for reliable reward collection, complementing prior methods that primarily focus on logit space. Additionally, recent work also adopts the pass@k training (Chen et al., 2025) that tolerates incorrect answer with potential valuable trajectory, which is orthogonal to this work and also a promising direction.

**Flow Matching in Reinforcement Learning.** As the most effective continuous modeling framework, flow matching (Lipman et al., 2022; Liu et al., 2022) is especially expert at generating high-dimensional signals, and has achieved remarkable success across a wide range of domains. Building on recent progress of RLVR, series of works have been proposed to further advance the generation quality in respective areas, including visual generation (Liu et al., 2025a; Li et al., 2025a; Xue et al., 2025) and robotics (Pfrommer et al., 2025; McAllister et al., 2025). These works leverage the flow as the policy model for optimization, which is distinct from RLFR that uses flow as environment for reward collection, and concentrates on velocity deviation metrics rather than reverse the process.

## 6 CONCLUSION

In this work, we analysis the auxiliary signals for reward shaping of RLVR from the perspective of latent space, and show that the latent space is highly expressive yet underexplored, complementing prior methods that focus closely on logit space. In light of this, RLFR offers a novel framework on shaping RLVR with flow reward,mwhere the flow field of model latent are constructed from either off-policy high-quality data and on-policy rejection sampling data, and the deviations of policy latents within it are quantified to serve as a reward signal, extending RLVR for latent reward uti-

lization. RLFR first demonstrates that a well-established flow field can be a sound environment for reward signal collection, yielding steady performance improvements across both language and multimodal benchmarks, highlighting the potential of latent substrate for reward design. Additionally, RLFR naturally leverages expert reasoning trajectories from off-policy data into the constitution of reward signal, instead of relying on self-confidence. Future directions involve scaling the flow environment to release the latent potential, and the prospect of latent signals for test-time scaling.

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

## A EXPERIMENTAL DETAILS

### A.1 SETTINGS

Experiments are conducted on 8 H20 GPUs. We use AdamW optimizer and sample 8 rollouts per prompt with (0.2, 0.28) clip range in policy loss for training. The prompt template is shown in Tab. 7, where we adopt bbox template for language training and tag template for multimodal training. In evaluation, we deploy a Qwen2.5-7B-Instruct model server for answer extraction and judge, and adopt DeepSeek v3.1 for more complex benchmarks, such as MathVision and MathVerse.

---

**RLFR training prompt**                                                **Bbox**

```
<|im_start|>system
Please reason step by step, and put your final answer within \\boxed{}.
<|im_end|>
<|im_start|>user
{{question}}<|im_end|>
<|im_start|>assistant
```

**RLFR training prompt**                                                **Tag**

```
<|im_start|>system
You should first thinks about the reasoning process in the mind and
then provides the user with the answer. Your answer must be in latex
format and wrapped in $...$. The reasoning process and answer are
enclosed within <think> </think> and <answer> </answer> tags,
respectively, i.e., <think> Since $1+1=2$, so the answer is $2$.
</think> <answer> $2$ </answer>, which means your output should start
with <think> and end with </answer>.
<|im_end|>
<|im_start|>user
{{question}}<|im_end|>
<|im_start|>assistant
```

Table 7: Training prompt for RLFR.

### A.2 TRAINING LOGS

We monitor the training dynamics of RLVR and RLFR in Fig. 6 for comparison. During training, the flow reward derived from latent space steadily improves the reasoning performance and accelerates the policy optimization, validating the reliability of the latent signals and their underexplored expressiveness. The policy entropy of RLFR also stabili zed at a slightly higher level during the training plateau compared to RLVR, underscoring the effectiveness of velocity deviation as the dense reward for encouraging exploration. While the response length shows healthy behavior with steady increases and adjustment, while no sign of degeneration are observed.

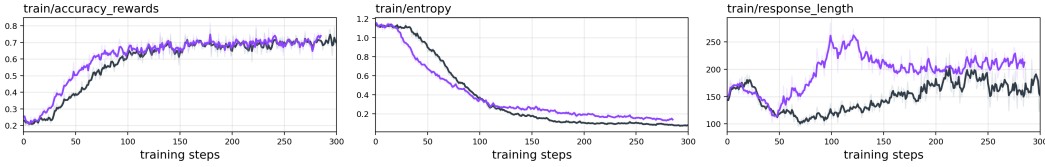

Figure 6: Training logs of **RLVR** and **RLFR** on Qwen2.5VL-3B.

## B EXTENDED ANALYSIS OF LATENT SPACE

Sec. 3.1 analysis the latent space of Qwen2.5-Base-7B at layer percentile of 0.5. We further show that the broad latent space exhibits the similar tendency throughout the LLM. In Fig. 7, we provide

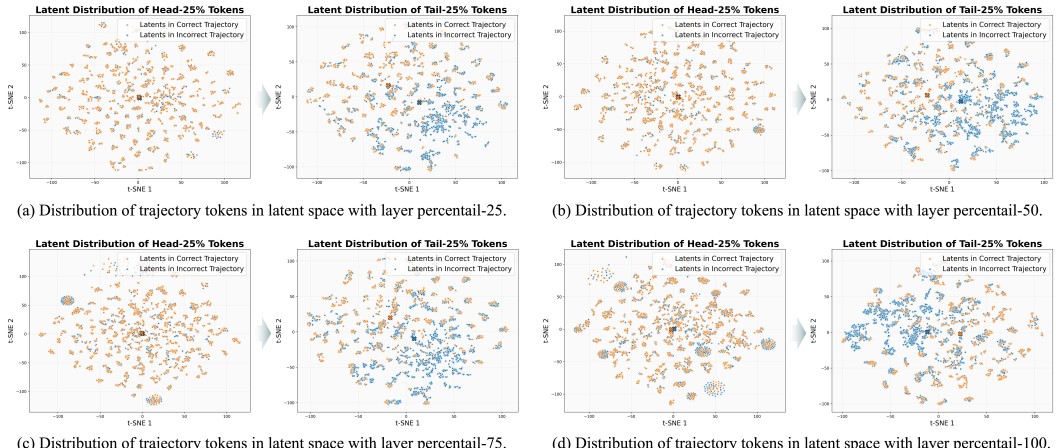

(a) Distribution of trajectory tokens in latent space with layer percentail-25.   (b) Distribution of trajectory tokens in latent space with layer percentail-50.

(c) Distribution of trajectory tokens in latent space with layer percentail-75.   (d) Distribution of trajectory tokens in latent space with layer percentail-100.

Figure 7: Distribution of reasoning trajectory tokens in latent space across different layer percentiles of the Qwen2.5-Base-7B, which consistently show expressive signals on tail trajectory tokens, highlighting the broader potential of latent space for reward signal collection.

the latent distributions of reasoning trajectory tokens at the $\{25, 50, 75, 100\}$ layer percentiles, and found that there is no evidence of contradiction and specialization across layer positions, instead, the latent space exhibits coherent and consistent signals for trajectory quality identification. In practice, we exclude the 100th percentile in training, as the last hidden states are heavily modulated by the $lm\_head$ for logit prediction, and we therefore rely on intermediate percentiles for reward collection.

## C  THEORETICAL ANALYSIS

### C.1  DERIVATION OF SCORE FUNCTION FOR VELOCITY

Here, we provide the derivations for Eq. 6 that establish the connection between velocity prediction and score function.

*Proof.* We consider the linear interpolation

$$\boldsymbol{x}_t = \alpha_t \boldsymbol{x}_0 + \beta_t \boldsymbol{x}_1, \qquad x_0 \sim \mathcal{N}(0, I), \ \ x_1 \sim p_{\text{data}},$$

where $\boldsymbol{x}_0$ and $\boldsymbol{x}_1$ are independent. We denote derivatives $\dot{\alpha}_t = \frac{d}{dt}\alpha_t$, $\dot{\beta}_t = \frac{d}{dt}\beta_t$. Conditioned on the data $\boldsymbol{x}_1$, the noisy variable $\boldsymbol{x}_t$ is Gaussian:

$$p_t(\boldsymbol{x}_t \mid \boldsymbol{x}_1) \ \sim \ \mathcal{N}\big(\beta_t \boldsymbol{x}_1, \ \alpha_t^2 I\big),$$

with conditional score

$$\nabla_x \log p(\boldsymbol{x}_t \mid \boldsymbol{x}_1) = -\frac{\boldsymbol{x}_t - \beta_t \boldsymbol{x}_1}{\alpha_t^2}. \tag{11}$$

By Fisher's identity, the marginal score given by

$$\boldsymbol{s}_t(x) = \nabla_x \log p_t(x) = \mathbb{E}\big[\nabla_x \log p(\boldsymbol{x}_t \mid \boldsymbol{x}_1) \mid \boldsymbol{x}_t = x\big]$$

$$= -\frac{1}{\alpha_t^2}\Big(x - \beta_t \mathbb{E}[\boldsymbol{x}_1 \mid \boldsymbol{x}_t = x]\Big). \tag{12}$$

Rearranging yields

$$\mathbb{E}[\boldsymbol{x}_1 \mid \boldsymbol{x}_t = x] = \frac{1}{\beta_t}\Big(x + \alpha_t^2 \boldsymbol{s}_t(x)\Big). \tag{13}$$

Considering $\frac{d}{dt}\boldsymbol{x}_t = \dot{\alpha}_t \boldsymbol{x}_0 + \dot{\beta}_t \boldsymbol{x}_1$, thus the velocity field is

$$\boldsymbol{v}_t(x) = \mathbb{E}\Big[\dot{\alpha}_t \boldsymbol{x}_0 + \dot{\beta}_t \boldsymbol{x}_1 \ \Big| \ \boldsymbol{x}_t = x\Big]$$

$$= \dot{\alpha}_t \mathbb{E}[\boldsymbol{x}_0 \mid \boldsymbol{x}_t = x] + \dot{\beta}\mathbb{E}[\boldsymbol{x}_1 \mid \boldsymbol{x}_t = x]. \tag{14}$$

Since $\boldsymbol{x}_0 = (\boldsymbol{x}_t - \beta_t \boldsymbol{x}_1)/\alpha_t$, we have

$$\mathbb{E}[\boldsymbol{x}_0 \mid \boldsymbol{x}_t = x] = \frac{x}{\alpha_t} - \frac{\beta_t}{\alpha_t} \mathbb{E}[\boldsymbol{x}_1 \mid \boldsymbol{x}_t = x]. \tag{15}$$

Substituting the Eq. 15 and Eq. 13, the velocity field is given by

$$\boldsymbol{v}_t(x) = \frac{\dot{\alpha}_t}{\alpha_t} x + \left( \dot{\beta}_t - \frac{\dot{\alpha}_t \beta_t}{\alpha_t} \right) \mathbb{E}[\boldsymbol{x}_1 \mid \boldsymbol{x}_t = x]$$

$$= \frac{\dot{\beta}_t}{\beta_t} x + \alpha_t^2 \left( \frac{\dot{\beta}_t}{\beta_t} - \frac{\dot{\alpha}_t}{\alpha_t} \right) \boldsymbol{s}_t(x). \tag{16}$$

For linear schedule, where $\alpha_t = 1 - t$ and $\beta_t = t$, we have

$$\boldsymbol{v}_t(x) = \frac{1}{t} x + \frac{1-t}{t} \boldsymbol{s}_t(x). \tag{17}$$

The score function is given by

$$\boldsymbol{s}_t(x) = -\frac{x}{1-t} + \frac{t}{1-t} \boldsymbol{v}_t(x). \tag{18}$$

Eq. 17 and Eq. 18 show the exact equivalence between the score function and velocity field under the linear interpolation schedule.

$\square$

## C.2 PROOF OF VARIATIONAL LOWER BOUND

Here, we provide the proof of Eq. 8 that establish the connection between the velocity deviations which measured by flow matching objective, and the likelihood given by evidence lower bound (ELBO) under target distribution. We indicate that the connection between ELBO and diffusion objective has been shown by previous works Kingma et al. (2021); Song et al. (2021); Kingma & Gao (2023), and we further extend it to velocity field.

*Proof.* For sample $\boldsymbol{y}$ that need to be evaluated for velocity deviation in reference flow distribution and consider the linear interpolation $\boldsymbol{y}_t = \alpha_t \boldsymbol{x}_0 + \beta_t \boldsymbol{y}$, with $\boldsymbol{x}_0 \sim p_{\text{init}}$. Let $\boldsymbol{s}_t(\boldsymbol{y}_t) = \nabla_y \log p_{\boldsymbol{v}_\phi}(\boldsymbol{y}_t)$ be the score of $\boldsymbol{y}_t$. Recall the flow matching objective in Eq. 5, we take

$$\boxed{\mathcal{L}_{\text{FM}}(\boldsymbol{y}; \phi) = \int_0^1 \mathbb{E}_{\boldsymbol{y} \sim q_t} \left[ \frac{1}{2} \| v_\phi(y, t) - u_t^y \|^2 \right] \mathrm{d}t.}$$

By continuity equation, the density and velocity of flow defined by $\boldsymbol{v}_\phi(\boldsymbol{y}_t, t)$ satisfies

$$\partial_t p_{\boldsymbol{v}_\phi}(\boldsymbol{y}_t) + \nabla \cdot \left( p_{\boldsymbol{v}_\phi}(\boldsymbol{y}_t) \, \boldsymbol{v}_\phi(\boldsymbol{y}_t, t) \right) = 0. \tag{19}$$

Given $\nabla \log p_{\boldsymbol{v}_\phi}(\boldsymbol{y}_t) = \nabla p_{\boldsymbol{v}_\phi}(\boldsymbol{y}_t) / p_{\boldsymbol{v}_\phi}(\boldsymbol{y}_t)$, we have

$$\partial_t \log p_{\boldsymbol{v}_\phi}(\boldsymbol{y}_t) = -\nabla \cdot \boldsymbol{v}_\phi(\boldsymbol{y}_t, t) - \boldsymbol{v}_\phi(\boldsymbol{y}_t, t) \cdot \nabla_y \log p_{\boldsymbol{v}_\phi}(\boldsymbol{y}_t). \tag{20}$$

Considering $\frac{d}{dt}\boldsymbol{y}_t = v_\phi(\boldsymbol{y}_t, t)$, and substituting the Eq. 20, the total derivative of $\log p_{\boldsymbol{v}_\phi}(\boldsymbol{y}_t)$ is

$$\frac{d}{dt} \log p_{\boldsymbol{v}_\phi}(\boldsymbol{y}_t) = \partial_t \log p_{\boldsymbol{v}_\phi}(\boldsymbol{y}_t) + \nabla_y \log p_{\boldsymbol{v}_\phi}(\boldsymbol{y}_t) \cdot \frac{d}{dt}\boldsymbol{y}_t$$

$$= -\nabla \cdot \boldsymbol{v}_\phi(\boldsymbol{y}_t, t) - \boldsymbol{v}_\phi(\boldsymbol{y}_t, t) \cdot \nabla_y \log p_{\boldsymbol{v}_\phi}(\boldsymbol{y}_t) + \nabla_y \log p_{\boldsymbol{v}_\phi}(\boldsymbol{y}_t) \cdot \frac{d}{dt}\boldsymbol{y}_t$$

$$= -\nabla \cdot \boldsymbol{v}_\phi(\boldsymbol{y}_t, t). \tag{21}$$

Therefore, integrating over $t \in [0, 1]$ yields the standard change-of-variables formula

$$\log p_{\boldsymbol{v}_\phi}(\boldsymbol{y}) = \log p_{\text{init}}(\boldsymbol{x}_0) - \int_0^1 \nabla \cdot \boldsymbol{v}_\phi(\boldsymbol{y}_t, t) \, dt$$

$$= \mathbb{E}_{x_0 \sim p_{\text{init}}}\big[\log p_{\text{init}}(\boldsymbol{x}_0)\big] - \int_0^1 \mathbb{E}_{y \sim p_{\text{data}}}\big[\nabla \cdot \boldsymbol{v}_\phi(\boldsymbol{y}_t, t)\big] \, dt$$

$$= C_0(x) - \int_0^1 \mathbb{E}_{y \sim p_{\text{data}}}\big[\langle \boldsymbol{v}_\phi(\boldsymbol{y}_t, t), \nabla \log p_{\text{data}}(\boldsymbol{y}_t)\rangle\big] \, dt \qquad (22)$$

where $\langle \cdot, \cdot \rangle$ denotes inner product, and we derive the last step from the Stein's identity. Considering

$$\langle a, g \rangle = \langle a, b \rangle - \tfrac{1}{2}\|b\|^2 + \tfrac{1}{2}\|g\|^2 - \tfrac{1}{2}\|g - b\|^2,$$

we thus substitute $a = \boldsymbol{v}_\phi(\boldsymbol{y}, t)$, $b = u_t^y$, and $g = \nabla \log p_{\text{data}}(y)$, yielding

$$\langle \boldsymbol{v}_\phi, \nabla \log p_{\text{data}} \rangle = \langle \boldsymbol{v}_\phi, u_t^y \rangle - \tfrac{1}{2}\|u_t^y\|^2 + \tfrac{1}{2}\|\nabla \log p_{\text{data}}\|^2 - \tfrac{1}{2}\|\nabla \log p_{\text{data}} - u_t^y\|^2.$$

We simplify $\boldsymbol{v}_\phi(y_t, t)$ as $\boldsymbol{v}_\phi$ and $p_{\text{data}}(\boldsymbol{y}_t)$ as $p_{\text{data}}$ for clarity, and have

$$\int_0^1 \mathbb{E}_{p_{\text{data}}}\big[\langle \boldsymbol{v}_\phi, \nabla \log p_{\text{data}} \rangle\big] \, dt = \int_0^1 \mathbb{E}_{p_{\text{data}}}\big[\langle \boldsymbol{v}_\phi, u_t^y \rangle\big] \, dt + B(y), \qquad (23)$$

where $B(y)$ depends only on the fixed bridge $(p_{\text{data}}, u_t^y)$, and have

$$B(y) = \int_0^1 \mathbb{E}_{p_{\text{data}}}\Big[ -\tfrac{1}{2}\|u_t^y\|^2 + \tfrac{1}{2}\|\nabla \log p_{\text{data}}\|^2 - \tfrac{1}{2}\|\nabla \log p_{\text{data}} - u_t^y\|^2\Big] \, dt. \qquad (24)$$

Considering the Fenchel–Young inequality

$$\langle \boldsymbol{v}_\phi, u_t^y \rangle \geq -\frac{1}{2\lambda}\|\boldsymbol{v}_\phi\|^2 - \frac{\lambda}{2}\|u_t^y\|^2, \qquad \lambda > 0,$$

where we substitute into Eq. 23, and havex

$$\int_0^1 \mathbb{E}_{p_{\text{data}}}\big[\langle \boldsymbol{v}_\phi, \nabla \log q_{\text{data}} \rangle\big] \, dt \geq \int_0^1 \mathbb{E}_{p_{\text{data}}}\Big[ -\frac{1}{2\lambda}\|\boldsymbol{v}_\phi\|^2 - \frac{\lambda}{2}\|u_t^y\|^2\Big] \, dt + B(y)$$

$$= \int_0^1 \mathbb{E}_{p_{\text{data}}}\Big[ -\frac{1}{2\lambda}\|\boldsymbol{v}_\phi - \lambda u_t^y\|^2\Big] \, dt + B(y) \qquad (25)$$

For the special case $\lambda = 1$, this simplifies to the equivalence given by Eq 8. Therefore, combining with Eq. 22, we obtain

$$\log p_{\boldsymbol{v}_\phi}(\boldsymbol{y}) \geq C(y) - \lambda \int_0^1 \mathbb{E}_{y \sim p_{\text{data}}}\Big[\tfrac{1}{2}\|\boldsymbol{v}_\phi(\boldsymbol{y}_y, t) - u_t^y\|^2\Big] \, dt,$$

which establish the connection between the velocity and likelihood under target distribution. $\qquad \square$

## D   CASE STUDY

**Question and Ground Truth**

**Question:** Jen enters a lottery by picking 4 distinct numbers from $S = \{1, 2, 3, \cdots, 9, 10\}$. 4 numbers are randomly chosen from $S$. She wins a prize if at least two of her numbers were 2 of the randomly chosen numbers, and wins the grand prize if all four of her numbers were the randomly chosen numbers. The probability of her winning the grand prize given that she won a prize is $\frac{m}{n}$ where $m$ and $n$ are relatively prime positive integers. Find $m + n$.
**Ground Truth:** 116

**Response from RLVR Baseline**

To solve this problem, we need to calculate the probability of Jen winning the grand prize given that she won a prize. We can break this down into several steps:

1. Calculate the total number of ways to choose 4 numbers from the set ( $S = \{1, 2, 3, \ldots, 10\}$ ).
2. Calculate the number of ways Jen can win a prize (i.e., at least 2 of her numbers are among the 4 chosen numbers).
3. Calculate the number of ways Jen can win the grand prize (i.e., all 4 of her numbers are among the 4 chosen numbers).
4. Use the conditional probability formula to find the probability of winning the grand prize given that she won a prize.

Let's go through these steps one by one.

1. The total number of ways to choose 4 numbers from 10 is given by the combination formula ( $\binom{10}{4}$ ).
2. The number of ways Jen can win a prize is the sum of the number of ways she can have exactly 2, 3, or 4 of her numbers among the 4 chosen numbers. This can be calculated as:
- Number of ways to have exactly 2 of her numbers among the 4 chosen numbers: ( $\binom{4}{2} \cdot \binom{6}{2}$ )
- Number of ways to have exactly 3 of her numbers among the 4 chosen numbers: ( $\binom{4}{3} \cdot \binom{6}{1}$ )
- Number of ways to have exactly 4 of her numbers among the 4 chosen numbers: ( $\binom{4}{4} \cdot \binom{6}{0}$ )
3. The number of ways Jen can win the grand prize is simply the number of ways to choose 4 numbers from her 4 chosen numbers, which is ( $\binom{4}{4} \cdot \binom{6}{0} = 1$ ).
4. The probability of winning the grand prize given that she won a prize is the ratio of the number of ways to win the grand prize to the number of ways to win a prize.

Let's calculate this using Python.

```python
import math

# Total number of ways to choose 4 numbers from 10
total_ways = math.comb(10, 4)

# Number of ways to win a prize
ways_to_win_prize = (math.comb(4, 2) * math.comb(6, 2)
                + math.comb(4, 3) * math.comb(6, 1)
                + math.comb(4, 4) * math.comb(6, 0))

# Number of ways to win the grand prize
ways_to_win_grand_prize = math.comb(4, 4) * math.comb(6, 0)

# Probability of winning the grand prize given that she
# won a prize
probability = ways_to_win_grand_prize / ways_to_win_prize

from fractions import Fraction
fraction = Fraction(ways_to_win_grand_prize, ways_to_win_prize)
m, n = fraction.numerator, fraction.denominator

result = m + n
print(result)

output 47
```

The probability of Jen winning the grand prize given that she won a prize is ( $\frac{1}{47}$ ).
Therefore, the values of (m) and (n) are 1 and 47, respectively. The sum (m + n) is (1 + 47 = 48).

So, the final answer is ( $\boxed{48}$ ).

**Response from RLFR**

To solve this problem, we need to calculate the probability of Jen winning the grand prize given that she won a prize. Let's break down the problem step-by-step:

1. **Total number of ways to choose 4 numbers from 10:** $\binom{10}{4} = \frac{10!}{4!(10-4)!} = 210$

2. **Number of ways Jen can win a prize:**
- Jen wins a prize if at least 2 of her numbers are among the 4 chosen numbers.
- We need to calculate the number of ways she can have 2, 3, or 4 of her numbers among the 4 chosen numbers.
- Number of ways to have exactly 2 of her numbers among the 4 chosen numbers: $\binom{4}{2} \cdot \binom{6}{2} = 6 \cdot 15 = 90$
- Number of ways to have exactly 3 of her numbers among the 4 chosen numbers: $\binom{4}{3} \cdot \binom{6}{1} = 4 \cdot 6 = 24$
- Number of ways to have exactly 4 of her numbers among the 4 chosen numbers: $\binom{4}{4} \cdot \binom{6}{0} = 1 \cdot 1 = 1$
- Total number of ways Jen can win a prize: $90 + 24 + 1 = 115$

3. **Number of ways Jen can win the grand prize:** - Jen wins the grand prize if all 4 of her numbers are among the 4 chosen numbers.
- There is only 1 way for this to happen: $\binom{4}{4} = 1$

4. **Probability of winning the grand prize given that she won a prize:** $\frac{\text{Number of ways to win the grand prize}}{\text{Number of ways to win a prize}} = \frac{1}{115}$

5. **Final answer:**
The probability is $\frac{1}{115}$, so $m = 1$ and $n = 115$. Since 1 and 115 are relatively prime, the final answer is: m + n = 1 + 115 = 116
The final answer is: $\boxed{116}$

# E  THE USE OF LARGE LANGUAGE MODELS

We used GPT5 solely for limited auxiliary purposes: (i) polishing the language of paragraphs, and (ii) generating refined format of latex code for showing. All scientific concepts, methodologies, and results were independently conceived, verified, and validated by the authors. Any outputs suggested by the LLM were manually reviewed to ensure correctness.

