# OpenReview forum: "RLFR: Extending Reinforcement Learning for LLMs with Flow Environment"
_ICLR.cc/2026/Conference — Submitted to ICLR 2026_

### Official Review · Reviewer_VU9d · 2025-10-27

**Soundness:** 1
**Presentation:** 2
**Contribution:** 2
**Rating:** 2
**Confidence:** 5

**Summary:**

This paper proposes RLFR, a reinforcement-learning framework for large language models that introduces flow rewards derived from latent representations. The method extends RL with verifiable rewards by constructing flow fields in the latent space from both off-policy expert data and on-policy rejection-sampling data. The deviation of policy latents from the reference flow field is quantified as a reward signal. Experiments on language and multimodal reasoning benchmarks show modest gains over RLVR and entropy-based shaping methods.

**Strengths:**

- Addresses a limitation of RLVR (coarse, binary reward signal) and explores a new direction of reward shaping.

- Attempts to leverage latent-space information rather than logit-space entropy, which is conceptually interesting.

- Includes relatively broad experiments across multiple model families and tasks.

- Implementation details and ablation studies are clearly reported.

**Weaknesses:**

- The flow-matching variable is borrowed from continuous diffusion literature but has no meaningful definition in the discrete LLM latent process. The “velocity deviation” term is therefore ill-posed; the claimed connection between velocity deviation and likelihood lacks grounding.


- LLM hidden states are conditionally dependent and layer-specific, not time-continuous trajectories. Treating them as samples evolving under a flow field appears ad hoc and unsupported by either theory or evidence.


- The flow reward effectively measures reconstruction error in a surrogate latent regression task, but it is unclear why this metric correlates with reasoning quality or exploration value.


- The reported performance gains are small (often 1-2%), without statistical tests or variance analysis to show significance. It is unclear whether improvements stem from flow rewards or simply additional fine-tuning.

- Key equations are overloaded with symbols whose roles shift across sections. The method description mixes notations from RL and flow matching, making the algorithm hard to reproduce or verify.

**Questions:**

What is the precise definition of the time variable in the context of LLM latent representations?

How does the velocity field interact with discrete transformer layers or token positions?

Is there empirical evidence that smaller “velocity deviations” correspond to higher-quality reasoning steps?

How sensitive are results to the choice of layer sampling, timestep, and reward scaling parameters?

Could the improvements be replicated if the same auxiliary loss were applied directly on latent reconstruction without RL?

---

> ### Author Response · Authors · 2025-11-27
> **Response to Reviewer VU9d (1/4)**
>
> We thank the reviewer for their feedback and positive assessment about interesting idea of our paper! We address specific concerns and questions below.
>
> > **W1: Clarification on flow matching in discrete LLM latent process and the grounding of connection between velocity deviation and likelihood.**
>
> Note that we do not define flow matching on discrete variables, and it may have a misunderstanding. We would like to note that **the latent hidden states of LLM are continuous**, which is exactly where the flow matching can be used for. And the velocity deviation is naturally to be defined on such continuous hidden states, just like it defined on other continuous variables, e.g., VAE latents for image modeling.
>
> The connection between diffusion objective and ELBO on log-likelihood has been well established by previous works [1][2], providing strong theoretically grounding, and we further establish the connection between flow matching objective (i.e., velocity deviation) with ELBO on log-likelihood in Appendix C.2, where the underlying principle is essentially the same.
>
> > **W2: LLM latents are not time-continuous trajectories, and treating them evolving under a flow field appears unsupported.**
>
> Notably, we’d like to note that **most continuous variables modeled by flow matching before is not time-continuous trajectories, but we add noise to make them time-continuous trajectories**. Here, we do the same thing, and add noise to LLM latents to model latents distribution, just like how we model image distribution with added noise.
>
> What different is that **we do not evolve the reverse process that sample from pure noise to clean data for generation**, but rather leverage the velocity prediction loss given by the forward process of flow network as a metric to evaluate whether current latents are within the data distribution formed by flow network. Basically, smaller velocity prediction loss indicates that current latents are within the flow distribution, while larger velocity prediction loss indicates the flow network is unfamiliar with current latents, and the latents are outside the flow distribution.
>
> We’d like to note that the core idea of this work is to using flow matching objective (i.e., velocity prediction) as a metric to excavate underexplored LLM latent space for reward utilization, and we do not use the reverse process that evolve the latents along the time trajectory for generation.
>
> > **W3: Why flow reward given by reconstruction error in a surrogate latent regression task correlating with reasoning quality.**
>
> That’s a great question, and we could naturally continue from the above question. As we first pretrain flow network on LLM latents with high quality CoT reasoning data for offline start, and continuously update the flow network with online rejection sampling data (filtered by answer correctness) in reinforcement learning, therefore, we could assume the latents’ distribution modeled by flow network is highly correlated with high quality reasoning trajectory, which serves as the target manifold that policy latents are expected to approach.
>
> In view of this, the velocity prediction loss could be a metric to evaluate how far current latents deviate from the flow distribution, and we use it as flow reward to encourage alignment with high-quality reasoning trajectory.
> One thing to note is that the latent is not simply a token-level embedding, but interacts with all preceding tokens through causal attention for context compression, which is a merit that we could not utilize in logit space.

---

> ### Author Response · Authors · 2025-11-27
> **Response to Reviewer VU9d (2/4)**
>
> > **W4: Concern about the performance improvement and whether improvements come from flow rewards or additional finetuning.**
>
> We would like to note that **we compare our method directly with RLVR, rather than the base model**. And we perform the same training steps on the same training data for comparison, and the only different is the flow reward in our method.
> Basically, 1-2 pp improvement over basic RLVR is relative adequate to validate the effectiveness of the method, and we list the performance improvement in recent reports, e.g., 1.7pp improvement in Entropy Adv. [3], 1.55 pp improvement for CPGD [4], 1.1 pp improvement for MMR1 [5].
> And we further provide results on stronger DeepSeek-R1-Distill-Qwen-7B and weaker Qwen2.5-7B to validate the reliability of the RLFR, where the improvement is noticeable, validating that RLFR is widely applicable to LLMs of various capabilities.
> |                             | AIME25        | AIME25        | AIME24        | AIME24        | AMC23         | AMC23         | MATH500       | MATH500       | OlympiadBench | OlympiadBench |
> |-----------------------------|---------------|---------------|---------------|---------------|----------------|----------------|----------------|----------------|----------------|----------------|
> |                             | Pass@32       | Pass@1        | Pass@32       | Pass@1        | Pass@32        | Pass@1         | Pass@32        | Pass@1         | Pass@32        | Pass@1         |
> | RLVR&nbsp;(Qwen2.5-7B)           | 23.3          | 3.3           | 26.7          | 13.3          | 72.2           | 40.9           | 90.6           | 73.4           | 60.6           | 36.3           |
> | RLFR&nbsp;(Qwen2.5-7B)           | 26.7          | 10.0          | 40.0          | 16.7          | 77.1           | 43.3           | 92.6           | 75.8           | 67.1           | 39.2           |
> | RLVR&nbsp;(R1-Distill-Qwen-7B)   | 53.3          | 33.3          | 76.6          | 40.0          | 91.6           | 74.7           | 98.4           | 90.4           | 76.3           | 55.5           |
> | RLFR&nbsp;(R1-Distill-Qwen-7B)   | 63.3          | 36.7          | 76.6          | 50.0          | 93.9           | 77.1           | 98.4           | 91.8           | 78.8           | 56.8           |
>
> Typically, pass@1 and pass@k metrics are reported in literatures to show the greedy decoding capability of LLMs and the upper performance bound of LLMs, where RLFR demonstrates improvement on both.
>
> > **W5: Mixed notations from RL and flow matching.**
>
> Thanks for pointing this out in details. We would like to further clarify our notations.
>
> In sec.2 preliminaries, we use $\dot{x}_t$ to denote latent of variable $x$ with added t-timestep noise to introduce flow matching, and use $a=(a_1,...,a_K)$ to denote response token sequence to introduce reinforcement learning.
>
> In sec.3, we combine them and use $\dot{a}_{k,t}$ to denote the latent hidden state of response token $a_k$ with t-timestep added noise. Basically, $x$ is a placeholder variable and instantiated with $a$ in our method.
>
> > **Q1: The definition of time variable in the context of LLM latent representation.**
>
> As we discussed before, we add noise to LLM latents to model latents distribution, which is just like how we model image distribution with added noise. And the time variable is defined the same as previous for how much noise we add to the latents.
>
> We would like to pointing out that the identified timestep debiasing method in sec. 3.2 and together with the results in Figure 5 further illustrate that **larger timesteps with less noisy are preferred for flow reward in velocity prediction evaluation**.
>
> > **Q2: How velocity field interact with transformer layers or token position.**
>
> We would like to further clarify this. We obtain the hidden states of the transformer layer through the hook mechanism in PyTorch, and after that, the hidden states could be sent into the flow network for velocity prediction evaluation to yield flow reward.
>
> We would like to note that the flow network is disabled in rollout phase, and only activated when computing logit probabilities for policy updates with one step model forward. Therefore, we could attain the whole sequence tokens for their hidden states, and the additional computation overhead is quite controllable, and basically on par with efficiency of RLVR.
>
> |                | RLVR  | RLFR  |
> |----------------|-------|-------|
> | Training time (h) | 12.38 | 12.62 |

---

> ### Author Response · Authors · 2025-11-27
> **Response to Reviewer VU9d (3/4)**
>
> > **Q3: Empirical evidence that smaller velocity deviations correspond to higher-quality reasoning steps.**
>
> We monitor the velocity deviation (VD) loss of correct and incorrect trajectories during training below, where we could observe that the **correct trajectories typically correspond to lower velocity deviations, while the incorrect trajectories are much higher**, verifying that the distribution modeled by flow network behaves is quite high quality and as expected. And we could therefore use velocity deviation as metric for yielding reward.
>
> |            | VD (mean) | VD (max) | VD (min) |
> |------------|-----------|----------|----------|
> | Correct    | 1.24      | 4.87     | 0.54     |
> | Incorrect  | 1.92      | 7.44     | 0.93     |
>
> We also provide the case study of frequently reward tokens with positive flow reward and negative flow reward in Table 3, where flow rewards prefer tokens that practically execute the question, rather than empty content such as connection tokens.
>
> > **Q4: How sensitive are results to the choice of layer sampling, timestep, and reward scaling parameters.**
>
> We provide the ablations of utilized timesteps in velocity prediction for flow reward, and the results suggest that the larger timesteps with less noisy are preferred for flow reward, which is align with our timesteps debiasing method in sec. 3.2.
>
> The layer percentile is not much sensitive to final results with basically 0.2 pp fluctuation in average score tested on Qwen2.5VL-3B with percentile of {0.25}, {0.5}, {0.75}, and we use the collection of them {0.25, 0.5, 0.75} in our experiments for slightly higher performance with negligible computation overhead.
>
> The flow reward scaling parameters $\beta$ is sensitive to final results, which could not too large to overwhelm the outcome reward, or too small that contributes with no effect. Basically, we set $\beta$ as 0.01 in our experiments, and $\beta$ = 0.1 will cause nearly 1 pp overall decrease in Qwen2.5VL-3B setting, and $\beta$=0.001 will cause 0.4 pp decrease.

---

> ### Author Response · Authors · 2025-11-27
> **Response to Reviewer VU9d (4/4)**
>
> > **Q5: Could the improvements be replicated if the same velocity loss were applied on latent without RL.**
>
> That’s an interesting question. However, as the process-level feedback, the velocity loss alone cannot work well, and the final outcome reward is still necessary to stimulate the training process. Basically, one could regard the flow reward as KL term for analogy, where KL can either be used inside the reward term, or can be used outside as loss term, however, purely applying KL without outcome reward may have no benefit in basic cases.
>
> |                       | AIME25        |        | AIME24        |        | AMC23         |         | MATH500       |       | Olympiad | |
> |-----------------------|---------------|---------------|---------------|---------------|----------------|----------------|----------------|----------------|----------------|----------------|
> |                       | Pass@32       | Pass@1        | Pass@32       | Pass@1        | Pass@32        | Pass@1         | Pass@32        | Pass@1         | Pass@32        | Pass@1         |
> | Velocity (w/o RL)     | 16.7          | 3.3           | 40.0          | 16.7          | 78.3           | 38.5           | 90.0           | 72.4           | 63.4           | 34.8           |
> | RLFR                  | 33.3          | 10.0          | 56.6          | 30.0          | 83.1           | 59.0           | 92.6           | 80.6           | 66.1           | 41.7           |
>
> Actually, we test the potential of RLFR for dealing with non-verifiable tasks, where no outcome rewards are given. Specifically, we follow the setting in RLMT [6], and use 6k SFT data for flow network offline start and 7.8k RL prompts for reinforcement finetuning, and evaluate the result on CreativeWritingV3 (CWv3) [7] and AlpacaEval2 (AE2) [8]. We found that RLFR indeed has the ability to deal with non-verifiable tasks with improved performance, compared to base model and even instruct model. Despite the unstable training process, which may need further accommodation, the verified potential of RLFR is encouraging, which is a promising direction for future research.
>
> |                         | CWv3 | AE2  |
> |-------------------------|------|------|
> | Qwen2.5-7B              | 39.1 | 4.5  |
> | Qwen2.5-7B-Instruct     | 49.8 | 37.1 |
> | RLFR-Qwen2.5-7B         | 52.4 | 46.8 |
>
> ---
>
> **Reference:**
>
> [1] Kingma, Diederik, et al. "Variational diffusion models." NeurIPS 2021.
>
> [2] Kingma, Diederik, et al. "Understanding diffusion objectives as the elbo with simple data augmentation." NeurIPS 2023.
>
> [3] Cheng, Daixuan, et al. "Reasoning with exploration: An entropy perspective." arXiv preprint 2025.
>
> [4] Liu, Zongkai, et al. "CPGD: Toward Stable Rule-based Reinforcement Learning for Language Models." arXiv preprint 2025.
>
> [5] Leng, Sicong, et al. "MMR1: Enhancing Multimodal Reasoning with Variance-Aware Sampling and Open Resources." arXiv preprint 2025.
>
> [6] Bhaskar, Adithya, et al. "Language models that think, chat better." arXiv preprint 2025.
>
> [7] Paech, Samuel J. "Eq-bench creative writing benchmark v3." 2025.
>
> [8] Dubois, et al. "Length-controlled alpacaeval: A simple debiasing of automatic evaluators." 2024.

---

### Official Review · Reviewer_6eD9 · 2025-10-31

**Soundness:** 2
**Presentation:** 3
**Contribution:** 2
**Rating:** 4
**Confidence:** 3

**Summary:**

This paper introduced RLFR, reinforcement learning with flow rewards. It is a new reward shaping to extend RLVR. Instead of relying on binary signals, RLFR constructs flow fields in model's latent space using both off-policy data and on-policy rejection-sampled data. The velocity deviation serves as the flow rewards which provides a continuous signal to guide reasoning trajectories. The experiments are conducted on Math and logic tasks, and multimodal tasks, which shows consistent improvements over RLVR.

**Strengths:**

Standard RLVR requires verifiable rewards which limit the use cases. This paper introduced a new perspective, leveraging flow environment to train models via RL. This offers a novel method to the community to further improve model's reasoning ability.

The paper is clearly written with logical flow. The authors also provides clear ablation studies, which comprehensively analyzed the contribution of key components.

**Weaknesses:**

1. The empirical improvement is limited. From Table 1 and Table 2, RLFR achieved limited improvement on many tasks. For example, for logic Avg. the improvement is only 0.1 which is within noise level. For Math Avg. the improvement is 2.4. This raises my concern about how effective this method is.

2. Can this method deal with non-verifiable rewards? For example, some tasks are difficult to verify, like long report generation. Is this method  more applicable to such tasks than RLVR?

3. Some typos and formatting issues:
- Line 193-194, analysis -> analyze
- Line 195-196, should use \citep instead of \citet. Similarly, there are many formatting issues with misused \citet.
- There are some overlaps in Figure 1 for numbers. For example, the rightmost 52.1 overlaps with 52.0.

**Questions:**

In Table 1, on AIME24 Pass@1, why there is a drop around 3.4 for RLFR compared with RLVR?

---

> ### Author Response · Authors · 2025-11-27
> **Response to Reviewer 6eD9 (1/2)**
>
> We thank the reviewer for their thoughtful feedback and positive assessment about novel perspective and clear written of our paper! We address specific concerns and questions below.
>
> > **W1: Concern about performance improvement and effectiveness of the propose method.**
>
> For multimodal settings where RLFR achieve 0.1pp improvement on logic benchmarks by Qwen2.5VL-3B, we’d like to note that the training data of RLFR is based on MMK12, where only math samples are included. **We report performance on logic benchmarks to access out-of-domain generalization of RLFR**. While the Qwen2.5VL-7B setting still achieve 1 pp improvement on this OOD setting. For in-distribution math benchmarks, RLFR achieves steady 2.4 pp improvement over basic RLVR, which is relatively noticeable. We list the performance improvement in recent reports, e.g., 1.55 pp average improvement for CPGD [1], 1.1 pp improvement for MMR1 [2]. Note that the improvement is compared against the RLVR under the same training set, rather than base model.
>
> For language setting, RLFR achieves 1.5 pp improvement on Qwen2.5-Math-7B setting. And we further provide results on stronger DeepSeek-R1-Distill-Qwen-7B and weaker Qwen2.5-7B to validate the effectiveness of the RLFR, where the improvement is noticeable, validating that RLFR is widely applicable to LLMs of various capabilities. And the consistent improvement for different model sizes, model families in the manuscript further verify the reliability of RLFR.
>
> |                             | AIME25        |        | AIME24        |        | AMC23         |         | MATH500       |       | Olympiad | |
> |-----------------------------|---------------|---------------|---------------|---------------|----------------|----------------|----------------|----------------|----------------|----------------|
> |                             | Pass@32       | Pass@1        | Pass@32       | Pass@1        | Pass@32        | Pass@1         | Pass@32        | Pass@1         | Pass@32        | Pass@1         |
> | RLVR&nbsp;(Qwen2.5-7B)           | 23.3          | 3.3           | 26.7          | 13.3          | 72.2           | 40.9           | 90.6           | 73.4           | 60.6           | 36.3           |
> | RLFR&nbsp;(Qwen2.5-7B)           | 26.7          | 10.0          | 40.0          | 16.7          | 77.1           | 43.3           | 92.6           | 75.8           | 67.1           | 39.2           |
> | RLVR&nbsp;(R1-Distill-Qwen-7B)   | 53.3          | 33.3          | 76.6          | 40.0          | 91.6           | 74.7           | 98.4           | 90.4           | 76.3           | 55.5           |
> | RLFR&nbsp;(R1-Distill-Qwen-7B)   | 63.3          | 36.7          | 76.6          | 50.0          | 93.9           | 77.1           | 98.4           | 91.8           | 78.8           | 56.8           |
>
> > **W2: Whether the proposed method can deal with non-verifiable tasks.**
>
> We have observed the potential of RLFR for dealing with non-verifiable tasks, which is a promising direction for future research. Specifically, we follow the setting in RLMT [3], and use 6k SFT data for flow network offline start and 7.8k RL prompts for reinforcement finetuning, and evaluate the result on CreativeWritingV3 (CWv3) [4] and AlpacaEval2 (AE2) [5]. We found that **RLFR indeed has the ability to deal with non-verifiable tasks with improved performance, compared to base model and even instruct model**. However, the training process is not stable, and will collapsed in 60 steps, which may need further accommodation.
>
> |                         | CWv3 | AE2  |
> |-------------------------|------|------|
> | Qwen2.5-7B              | 39.1 | 4.5  |
> | Qwen2.5-7B-Instruct     | 49.8 | 37.1 |
> | RLFR-Qwen2.5-7B         | 52.4 | 46.8 |
> | RLMT-Qwen2.5-7B         | 65.7 | 54.0 |
>
> > **W3: Some typos.**
>
> We thank the reviewer for pointing this out in detail. We have updated them including citation format and Figure 1 for clear presentation in revision.

---

> ### Author Response · Authors · 2025-11-27
> **Response to Reviewer 6eD9 (2/2)**
>
> > **Q1: The performance drop around 3.4 on AIME24 Pass@1 for Qwen2.5-Math-1.5B.**
>
> We would like to note that we basically focus on the overall performance, where RLFR achieves 1.28 pp improvement on Pass@1 and 2.43 pp improvement on Pass@32 for Qwen2.5-Math-1.5B. Considering there are only 30 problems in AIME24 with high difficulty, and one problem failed will lead to 3.4 (1/30) pp decrease in performance, we further **provide avg@k results for more robust evaluation, and RLFR achieves steady 1.53 pp improvements in this setting**.
>
> |        | AIME25        |        | AIME24        |        | AMC23         |         | MATH500       |       | Olympiad | |
> |--------|---------------|---------------|---------------|---------------|----------------|----------------|----------------|----------------|----------------|----------------|
> |        | Pass@32       | Avg@32        | Pass@32       | Avg@32        | Pass@32        | Avg@32         | Pass@32        | Avg@32         | Pass@32        | Avg@32         |
> | RLVR   | 23.3          | 4.6           | 46.7          | 15.8          | 81.9           | 41.3           | 91.8           | 71.3           | 62.7           | 33.8           |
> | RLFR   | 30.0          | 5.9           | 50.0          | 15.5          | 83.1           | 44.9           | 92.4           | 72.6           | 63.1           | 35.6           |
>
> ---
> **Reference:**
>
> [1] Liu, Zongkai, et al. "CPGD: Toward Stable Rule-based Reinforcement Learning for Language Models." arXiv preprint 2025.
>
> [2] Leng, Sicong, et al. "MMR1: Enhancing Multimodal Reasoning with Variance-Aware Sampling and Open Resources." arXiv preprint 2025.
>
> [3] Bhaskar, Adithya, et al. "Language models that think, chat better." arXiv preprint 2025.
>
> [4] Paech, Samuel J. "Eq-bench creative writing benchmark v3." 2025.
>
> [5] Dubois, et al. "Length-controlled alpacaeval: A simple debiasing of automatic evaluators." 2024.

---

### Official Review · Reviewer_A3g4 · 2025-11-01

**Soundness:** 3
**Presentation:** 3
**Contribution:** 3
**Rating:** 6
**Confidence:** 2

**Summary:**

This paper proposes RLFR, a novel reward shaping framework to extend Reinforcement Learning with Verifiable Rewards (RLVR) for LLMs. The authors argue that standard RLVR with binary verification overlooks valuable exploration, while existing auxiliary signals from the logit space (like entropy) are limited. RLFR introduces a "flow reward" derived from the model's latent space. This method constructs a flow field using off-policy high-quality data and on-policy rejection sampling. The reward signal is then calculated as the "velocity deviation" of the policy's current latent states within this flow field, penalizing drift from the reference distribution. The central claim is that the latent space is an underexplored but highly expressive substrate for reward collection and that this flow-based reward provides a stable signal to improve LLM reasoning.

**Strengths:**

1. The primary strength is the novel use of flow matching and velocity deviation from the latent space to define a reward signal. This is a creative departure from common approaches that focus on outcome verification or logit-space signals (e.g., entropy, likelihood)

2. he method demonstrates consistent and robust performance gains. RLFR outperforms the base model, the standard RLVR baseline, and the Entropy Advantage (logit-space) baseline across a suite of six language and multimodal reasoning benchmarks (Figure 1). This holds true in detailed tables for both language (Table 1) and multimodal (Table 2)  tasks.

3. The authors provide a comprehensive set of ablations that validate the key components of the RLFR framework. The studies confirm the necessity of the offline start, online rejection sampling, and fluctuation filtering (Table 3). Furthermore, the ablations justify specific design choices, such as using outcome correctness as the rejection metric (Table 4) and the superiority of the proposed timestep debiasing method (Figure 4).

4. The paper provides good qualitative analysis. Figure 3 offers a clear visual motivation, suggesting that latent space distributions (especially for tail tokens) provide a more distinguishable signal for correctness than logit distributions.

**Weaknesses:**

1. The proposed method is highly complex, which may be its greatest drawback. Unlike a simple entropy bonus, RLFR requires training a separate flow network. This network must first be pre-trained on offline data and then continuously updated online using a rejection sampling buffer. This dual-optimization loop introduces substantial computational overhead and implementation challenges not present in the baselines.

2. The "Entropy Adv." baseline, used as a key comparison for logit-space shaping, performs poorly. In Figure 1, it underperforms the standard RLVR baseline in three of the six benchmarks. This is also seen in Table 1 for the Qwen2.5-Math-7B model. This suggests the entropy baseline may be weakly implemented or inherently flawed, making RLFR's outperformance of it less significant than its gains over the stronger RLVR baseline.

3. The paper is not perfectly clear about how latents from multiple layers are processed. It states latents are extracted from percentiles {0.25, 0.5, 0.75} and fed into a 4- or 6-layer flow network. However, the exact mechanism for combining these multi-layer features (e.g., concatenation, averaging) and integrating the "layer position embedding"  is not detailed, leaving a gap in reproducibility.

4. The repeated use of "Flow Environment"  is imprecise. In RL, the environment is the process being optimized (the LLM's decoding). The flow network is a component of the reward function, not a new environment. This terminological choice, while evocative, slightly obscures the method's actual mechanics.

**Questions:**

none

---

> ### Comment · Reviewer_A3g4 · 2025-11-21
>
> I still find the methodology in this paper quite novel, and I hope the author will address my concerns. I may consider raising my score based on the author's response.

---

> > ### Comment · Reviewer_A3g4 · 2025-11-26
> >
> > I hope the author could at least respond a bit to the reviews.

---

> ### Author Response · Authors · 2025-11-27
> **Response to Reviewer A3g4 (1/2)**
>
> We thank the reviewer for their thoughtful feedback and positive assessment about novelty and creativity of our paper! And we also sincerely thank the reviewer for their patience during our response preparation. We address specific concerns and questions below.
>
> > **W1: Computational and implementation complexity of the proposed method.**
>
> We compare the computation complexity of RLFR with basic RLVR below, where RLFR exhibits negligible computation overhead, and the detailed training profile of RLFR further show that the forward and backward update of flow network only occupied 2.3% of the one-step training time. We’d like to note that the rollout still dominates the overall runtime, outweighing the cost of policy update, and the flow update only constitutes a small fraction of policy update, which is quite controllable. Additionally, the offline start of flow network often can be done within 1 hour for reasonable initialization.
>
> |                | RLVR  | RLFR  |
> |----------------|-------|-------|
> | Training time (h) | 12.38 | 12.62 |
>
> |                     | Rollout | Policy update / full batch | Policy update / full batch (64 mini steps) LLM forward + LLM backward | Policy update / full batch (64 mini steps) Flow forward + Flow backward |
> |---------------------|------------|-----------------------------|----------------------------------------------------------------------------|------------------------------------------------------------------------------|
> | Time&nbsp;(s)            | 102&nbsp;(53.1%) | 86 (44.7%)               | 16.64 + 44.16 (31.6%)                                                     | 1.28 + 3.2 (2.3%)
>
> The implementation of RLFR is straightforward. Actually, RLFR can be viewed as a lightweight actor-critic architecture, with the flow network serves as the role of value model, where the online updating of the flow network could integrate naturally into current value model implementation for convenience.
>
> > **W2: The weak performance of Entropy Adv. baseline.**
>
> We implement the Entropy Adv. baseline based on the pytorch code provided in their report, as the work haven’t open sourced yet, and rerun the experiment with more training steps (from 200 steps to 800 steps), leading to slight performance increase, but still fell short to ideal results. We observe that the Entropy Adv. performs 2000 steps training in their report, which is prohibitively a huge cost in our experiments and introduces an unfair comparison.
>
> |                     | AIME25        |        | AIME24        |        | AMC23         |         | MATH500        |        | Olympiad |  |
> |---------------------|---------------|---------------|---------------|---------------|----------------|----------------|-----------------|-----------------|----------------|----------------|
> |                     | Pass@32       | Pass@1        | Pass@32       | Pass@1        | Pass@32        | Pass@1         | Pass@32         | Pass@1         | Pass@32        | Pass@1         |
> | Entropy Adv.        | 26.6          | 6.6           | 53.3          | 20.0          | 75.9           | 54.2           | 93.0            | 75.6           | 61.2           | 36.4           |
> | Entropy Adv. (rerun)| 30.0          | 10.0          | 50.0          | 23.3          | 78.3           | 56.6           | 93.4            | 76.2           | 61.9           | 37.2           |
>
> We further include PRIME [1] and ReasonFlux-PRM [2] for baseline comparison. The PRIME is a token-level process reward approach, which is relative to RLFR, and ReasonFlux-PRM integrates additional process reward model (e.g., Qwen-2.5-Math-PRM-7B) for step-level reward yielding. We observe that RLFR even achieving on par performance with ReasonFlux-PRM, while with noticeable reduced training cost (12.62h for RLFR; 21.53h for ReasonFlux-PRM).
>
> | Method          | AIME25        |        | AIME24        |        | AMC23         |         | MATH500        |        | Olympiad  |  |
> |-----------------|---------------|---------------|---------------|---------------|----------------|----------------|-----------------|-----------------|----------------|----------------|
> |                 | Pass@32       | Pass@1        | Pass@32       | Pass@1        | Pass@32        | Pass@1         | Pass@32         | Pass@1         | Pass@32       | Pass@1        |
> | PRIME           | 33.3          | 16.7          | 53.3          | 16.7          | 79.5           | 55.4           | 91.8            | 76.2           | 65.3          | 37.9          |
> | ReasonFlux-PRM  | 36.7          | 13.3          | 56.6          | 26.7          | 81.9           | 57.8           | 92.8            | 80.4           | 65.7          | 41.4          |
> | RLFR            | 33.3          | 10.0          | 56.6          | 30.0          | 83.1           | 59.0           | 92.6            | 80.6           | 66.1          | 41.7          |
>
> The added baseline results are included in the manuscripts (Table. 1) for strengthen.

---

> ### Author Response · Authors · 2025-11-27
> **Response to Reviewer A3g4 (2/2)**
>
> > **W3: How latents from multiple layers are processed, especially how to combine and layer position embedding.**
>
> We would like to further clarify this clearly. Note that the flow network is disabled in rollout phase, and only activated when computing logit probabilities for policy updates. In this one step model forward, the latents from multiple layer percentiles (e.g., 3) are extracted, and we calculate the velocity deviation through flow network for each layer percentile latent, and average them for final velocity deviation of this token, thus yielding the flow reward.
>
> The layer position embedding is to indicate the flow network the layer percentile of the passed latent, and implemented the same as the time embedding in flow architecture. In practical, we add the layer position embedding, time embedding, and the conditional embedding (as discussed in Table 6) to serve as global embedding to modulate the hidden states in flow network.
>
> ---
> **Reference:**
>
> [1] Cui, Ganqu, et al. "Process reinforcement through implicit rewards." arXiv preprint 2025.
>
> [2] Zou, Jiaru, et al. "ReasonFlux-PRM: Trajectory-Aware PRMs for Long Chain-of-Thought Reasoning in LLMs." arXiv preprint 2025.

---

> ### Comment · Reviewer_A3g4 · 2025-11-28
>
> Thank you for your reply. I believe some of my questions have been resolved, so I have decided to increase my rating. However, it seems that the system has currently locked the option to edit the rating, so my correct rating should be 8.

---

### Official Review · Reviewer_x842 · 2025-11-01

**Soundness:** 3
**Presentation:** 3
**Contribution:** 3
**Rating:** 4
**Confidence:** 3

**Summary:**

This paper introduces RLFR (reinforcement learning with flow rewards), a novel framework that enhances LLMs' reasoning capabilities by moving beyond the limitations of binary-reward reinforcement learning. The core innovation is the use of flow matching in the model's latent space to generate dense, process-oriented reward signals. Specifically, RLFR constructs a flow field from high-quality data and quantifies the velocity deviation of the policy's latent states within this field. More minor deviations are rewarded, while larger deviations are penalized. The authors theoretically connect velocity deviation to the evidence lower bound of log-likelihood, establishing a foundation for this reward signal.

**Strengths:**

The idea of using flow matching to generate a continuous reward signal from latent-space dynamics is interesting. While flow matching is established in generative modeling, its application as an environment for quantifying the quality of intermediate reasoning steps in RL is a fresh and compelling combination of ideas. It effectively addresses a known weakness of RLVR (sparse, binary rewards) with a technically sophisticated solution. The experiments on language and multimodal reasoning benchmarks demonstrate that RLFR consistently outperforms baseline RLVR and entropy-based shaping methods, showing more stable training and improved performance.

**Weaknesses:**

The primary baselines are basic RLVR and an entropy-based shaping method. The paper would be significantly strengthened by comparing against more competitive baselines. The authors note PRMs' high cost as a bottleneck, but it would be valuable to see a direct comparison, even on a smaller scale, to show whether RLFR can be a lower-cost alternative that achieves good performance. Compared to the baselines, the proposed method shows limited improvements. Further detailed experimental analysis is needed to show the reliability of the proposed method.

**Questions:**

Please refer to "Weaknesses".

---

> ### Author Response · Authors · 2025-11-27
> **Response to Reviewer x842 (1/2)**
>
> We thank the reviewer for their thoughtful feedback and positive assessment about fresh and interesting idea of our paper! We address specific concerns and questions below.
>
> > **W1: Comparing against more competitive baselines.**
>
> The most relative works to RLFR is token-level process reward approach, including SGPO [2], GTPO [3], which haven’t open sourced yet, we therefore include the PRIME [4] for comparison. Additionally, we notice that the comparable baselines in this line of works are relatively limited, e.g., only RLVR in [1] and [3], RLVR and PRIME in [2]. In view of this, we further add Qwen-2.5-Math-PRM-7B as step-level PRM baseline with implementation of ReasonFlux-PRM [5] for comparison, and RLFR achieves even 0.34 pp higher than this strong baseline, considering the huge disparity in training cost, which will be detailed next. The added baseline results are included in the manuscripts (Table. 1) for strengthen.
>
> | Method          | AIME25        |        | AIME24        |        | AMC23         |         | MATH500        |        | Olympiad  |  |
> |-----------------|---------------|---------------|---------------|---------------|----------------|----------------|-----------------|-----------------|----------------|----------------|
> |                 | Pass@32       | Pass@1        | Pass@32       | Pass@1        | Pass@32        | Pass@1         | Pass@32         | Pass@1         | Pass@32       | Pass@1        |
> | PRIME           | 33.3          | 16.7          | 53.3          | 16.7          | 79.5           | 55.4           | 91.8            | 76.2           | 65.3          | 37.9          |
> | ReasonFlux-PRM  | 36.7          | 13.3          | 56.6          | 26.7          | 81.9           | 57.8           | 92.8            | 80.4           | 65.7          | 41.4          |
> | RLFR            | 33.3          | 10.0          | 56.6          | 30.0          | 83.1           | 59.0           | 92.6            | 80.6           | 66.1          | 41.7          |
>
> > **W2: Direct comparison with PRM in computation efficiency.**
>
> We show the direct comparison with above ReasonFlux-PRM baseline that employed with Qwen-2.5-Math-PRM-7B in computation efficiency below, where RLFR exhibits on par efficiency with RLVR, while PRM shows non-negligible overhead. Note that the implementation of ReasonFlux-PRM is quite clean for combing process reward and outcome reward, and there is no other additional operation.
>
> |                | RLVR  | w/ PRM | RLFR  |
> |----------------|-------|--------------------------|-------|
> | Training time (h) | 12.38 | 21.53                   | 12.62 |
>
> We further provide the training profile of RLFR below, which can be observed that the computation of flow network only occupied 2.3% of the one-step training time.
>
> |                     | Rollout | Policy update / full batch | Policy update / full batch (64 mini steps) / LLM forward + LLM backward | Policy update / full batch (64 mini steps) / Flow forward + Flow backward |
> |---------------------|------------|-----------------------------|----------------------------------------------------------------------------|------------------------------------------------------------------------------|
> | Time&nbsp;(s)            | 102&nbsp;(53.1%) | 86 (44.7%)               | 16.64 + 44.16 (31.6%)                                                     | 1.28 + 3.2 (2.3%)                                                             |

---

> ### Author Response · Authors · 2025-11-27
> **Response to Reviewer x842 (2/2)**
>
> > **W3: More experimental results to show the reliability of the proposed method.**
>
> We further select stronger DeepSeek-R1-Distill-Qwen-7B and weaker Qwen2.5-7B to validate the reliability of the RLFR. The noticeable results further validate that RLFR is widely applicable to LLMs of various capabilities. And the consistent improvements over RLVR on different model size, model family, and multimodal setting provided in the manuscript further validate the reliability of the RLFR.
>
> |                             | AIME25        |        | AIME24        |        | AMC23         |         | MATH500       |       | Olympiad | |
> |-----------------------------|---------------|---------------|---------------|---------------|----------------|----------------|----------------|----------------|----------------|----------------|
> |                             | Pass@32       | Pass@1        | Pass@32       | Pass@1        | Pass@32        | Pass@1         | Pass@32        | Pass@1         | Pass@32        | Pass@1         |
> | RLVR&nbsp;(Qwen2.5-7B)           | 23.3          | 3.3           | 26.7          | 13.3          | 72.2           | 40.9           | 90.6           | 73.4           | 60.6           | 36.3           |
> | RLFR&nbsp;(Qwen2.5-7B)           | 26.7          | 10.0          | 40.0          | 16.7          | 77.1           | 43.3           | 92.6           | 75.8           | 67.1           | 39.2           |
> | RLVR&nbsp;(R1-Distill-Qwen-7B)   | 53.3          | 33.3          | 76.6          | 40.0          | 91.6           | 74.7           | 98.4           | 90.4           | 76.3           | 55.5           |
> | RLFR&nbsp;(R1-Distill-Qwen-7B)   | 63.3          | 36.7          | 76.6          | 50.0          | 93.9           | 77.1           | 98.4           | 91.8           | 78.8           | 56.8           |
>
> ---
> **Reference：**
>
> [1] Cheng, Daixuan, et al. “Reasoning with exploration: An entropy perspective.” arXiv preprint 2025.
>
> [2] Fei, Wu, et al. “Self-Guided Process Reward Optimization with Redefined Step-wise Advantage for Process Reinforcement Learning.” arXiv preprint 2025.
>
> [3] Tan, Hongze, et al. “Gtpo and grpo-s: Token and sequence-level reward shaping with policy entropy.” arXiv preprint 2025.
>
> [4] Cui, Ganqu, et al. “Process reinforcement through implicit rewards.” arXiv preprint 2025.
>
> [5] Zou, Jiaru, et al. “ReasonFlux-PRM: Trajectory-Aware PRMs for Long Chain-of-Thought Reasoning in LLMs.” arXiv preprint 2025.

---

### Author Response · Authors · 2025-12-03
**Concise Summary of Author Response**

We sincerely thank all the reviewers for their thoughtful suggestions. In the following, we present a concise summary of the key concerns and our corresponding responses.

> **The recognized contribution of this work. (Reviewer x842@s; Reviewer A3g4@s2; Reviewer 6eD9@s)**

Moving beyond binary verification for LLM reasoning, we realized that the expressive latent space is much underexplored for reward utilization, and demonstrate that a well-established flow field can be a sound environment for reward signal collection. We'd like to highlight that this is the first time the flow network is not employed with reverse sampling for data generation, but serve as a distribution metric to quantify the velocity deviations of policy latents for yielding reward signal.
**Reviewers x842, A3g4,** and **6eD9** recognized the interesting idea of our work.

> **Comparison with Process Reward Models (PRM) for computation efficiency. (Reviewer x842@w2; Reviewer A3g4@w1)**

We present the training time comparison, and show that RLFR (12.62h) exhibits on par efficiency with RLVR (12.38h), while PRM shows non-negligible overhead (21.53h). We further add this strong baseline to our comparison, that adopts Qwen-2.5-Math-PRM-7B with the implementation of ReasonFlux-PRM (quite clean for combing process reward and outcome reward with no other additional operation). And RLFR achieves even 0.34 pp higher, consideirng the huge reduced training cost.

> **More experiments to show the reliability. (Reviewer x842@w3; Reviewer 6eD9@w1; VU9d@w1)**

We further include stronger DeepSeek-R1-Distill-Qwen-7B and weaker Qwen2.5-7B for verification, and RLFR achieves 3.69 pp and 3.56 pp improvements compared to basic RLVR. And the consistent improvement for different model sizes (1.5B/3B/7B/8B), model families (Qwen/Llama), and multimodal settings in the manuscript further support the reliability of RLFR.

> **Whether the proposed method can deal with non-verifiable tasks. (Reviewer 6eD9@w2; VU9d@Q5)**

We follow the setting in RLMT, and use 6k SFT data for flow network offline start and 7.8k RL prompts for reinforcement finetuning, and evaluate the result on CreativeWritingV3 (CWv3) and AlpacaEval2 (AE2). We found that RLFR has the potential to deal with non-verifiable tasks with only flow reward for improved performance (52.4/46.8), compared to the base model (39.1/4.5) and even instruct model (49.8/37.1). However, the training process will collapsed in 60 steps, which may need further accommodation for stablization.

> **Some misunderstandings. (Reviewer VU9d@w1&w2@Q1)**

**Reviewer VU9d** suppose (1) our flow matching is defined on discrete LLM latents, which is unsupported by either theory or evidence. We clarify that the latent hidden states of LLM are continuous, which is exactly where the flow matching can be used for. (2) LLM latents are not time-continuous trajectories. We clarify that the time-continuous trajectories is modeled by progressively added noise in flow matching framework, rather than the variable itself. We provided point-by-point responses to **Reviewer VU9d** for clarify misunderstandings of our work, and we hope it can address the concerns of the reviewer.

Currently, Reviewer A3g4 has decided to raise score from 6 to 8, and no other updates for further comments.

We thank all the reviewers and AC for their time and dedication. We hope this concise summary could help AC quickly grasp the current status of this submission.

Best Regards,

Authors of Submissions 2717

---

### Meta-Review · Area_Chair_dd92 · 2025-12-28

**Summary:**

This work proposes to use flow matching in the model’s latent space to produce dense, process-oriented flow reward signals, and introduces the framework named RLFR, reinforcement learning with flow rewards, to overcome the shortcomings of the binary verification methods and the auxiliary signal-based methods. The authors claim that the flow field constructed by RLFR can be used to quantify the velocity deviation of the policy’s latent states, which serves as a continuous reward signal to guide reasoning trajectories. Experiments are conducted on both language and multimodal reasoning benchmarks to evaluate the performance of RLFR.

All reviewers acknowledge that the perspective from flow matching for RLVR is novel and interesting, and may become a new direction of reward shaping. Despite the strengths, the reviewers also share some similar concerns:

(1)	Three reviewers simultaneously have concerns about the limited performance improvements of RLFR. The other reviewer (A3g4) also conveyed an issue about baseline performance that may impede the outperformance of the proposed method. Similarly, Reviewer x842 pointed out that the paper would be significantly strengthened by including more competitive baselines.

(2)	Both reviewers x842 and A3g4 questioned the complexity and computational cost of the proposed framework, and suggested to compare the cost with the baselines.

(3)	Both reviewers A3g4 and VU9d noted that the adopted terms “flow matching” or “flow environment” are not precise or grounded.

(4)	Multiple reviewers also suggested that more implementation details are needed.

**Reviewer Concerns:**

During rebuttal, the authors included more experiment to compare with more competitive baselines, reported the computational cost, and provided results on more models to show noticeable improvements. The AC reviewed the paper and all the discussion process, and acknowledged that some of the questions were addressed, such as the implementation details and efficiency evaluation. However, the paper still needs to be strengthened in several aspects, including but not limited to more convincing experimental results to demonstrate superiority in terms of performance and efficiency, and more detailed explanations to support the terminology choice.

**Reviewer Scores:**

If reviewers had been able to participate fully in the discussion, Reviewer A3g4 would change their score to 8 as they acknowledged in the comment. The final scores would be 2, 4, 4, 8.

---

### Decision · Program_Chairs · 2026-01-26

Reject